# Single-cell and spatial analysis reveal interaction of *FAP*[+] fibroblasts and *SPP1*[+] macrophages in colorectal cancer

Jingjing Qi[1,2,3], Hongxiang Sun[1,2], Yao Zhang[4], Zhengting Wang[4], Zhenzhen Xun[1], Ziyi Li[1], Xinyu Ding[1], Rujuan Bao[1], Liwen Hong[4], Wenqing Jia[1], Fei Fang[1], Hongzhi Liu[1,2], Lei Chen [1], Jie Zhong[4], Duowu Zou[4], Lianxin Liu [5], Leng Han [6], Florent Ginhoux [1,7], Yingbin Liu[3], Youqiong Ye [1,2,4✉] & Bing Su [1,2,4✉]

Colorectal cancer (CRC) is among the most common malignancies with limited treatments other than surgery. The tumor microenvironment (TME) profiling enables the discovery of potential therapeutic targets. Here, we profile 54,103 cells from tumor and adjacent tissues to characterize cellular composition and elucidate the potential origin and regulation of tumor-enriched cell types in CRC. We demonstrate that the tumor-specific *FAP*[+] fibroblasts and *SPP1*[+] macrophages were positively correlated in 14 independent CRC cohorts containing 2550 samples and validate their close localization by immuno-fluorescent staining and spatial transcriptomics. This interaction might be regulated by chemerin, TGF-β, and interleukin-1, which would stimulate the formation of immune-excluded desmoplasic structure and limit the T cell infiltration. Furthermore, we find patients with high *FAP* or *SPP1* expression achieved less therapeutic benefit from an anti-PD-L1 therapy cohort. Our results provide a potential therapeutic strategy by disrupting *FAP*[+] fibroblasts and *SPP1*[+] macrophages interaction to improve immunotherapy.

[1] Shanghai Institute of Immunology, Department of Immunology and Microbiology, and the Ministry of Education Key Laboratory of Cell Death and Differentiation, Shanghai Jiao Tong University School of Medicine, Shanghai, China. [2] Shanghai Jiao Tong University School of Medicine-Yale Institute for Immune Metabolism, Shanghai Jiao Tong University School of Medicine, Shanghai, China. [3] Department of Biliary and Pancreatic Surgery, Renji Hospital, Shanghai Jiao Tong University School of Medicine, Shanghai, China. [4] Department of Gastroenterology, Center for Immune-related Diseases, Ruijin Hospital, Shanghai Jiao Tong University School of Medicine, Shanghai, China. [5] Department of Hepatobiliary Surgery, Anhui Province Key Laboratory of Hepatopancreatobiliary Surgery, The First Affiliated Hospital of USTC, Division of Life Sciences and Medicine, University of Science and Technology of China, Hefei 230001, China. [6] Department of Biochemistry and Molecular Biology, The University of Texas Health Science Center at Houston McGovern Medical School, Houston, TX 77030, USA. [7] Singapore Immunology Network (SIgN), A*STAR, 8A Biomedical Grove, Immunos Building, Level 3 and 4, Singapore 138648, Singapore. [8] These authors jointly supervised this work: Youqiong Ye, Bing Su. ✉email: youqiong.ye@shsmu.edu.cn; bingsu@sjtu.edu.cn

Colorectal cancer (CRC) is the third most common malignancy (after lung and breast cancer) and causes around 800,000 deaths annually world-wide. Recently, following its success in previously difficult-to-treat solid tumors, such as melanoma and lung cancer, an immune checkpoint blockade (ICB) strategy was applied to CRC treatment[1]. There have been many studies exploring T cells for effective anti-tumor immunotherapies[2]. However, the PD-1-targeting antibody, pembrolizumab, is only effective for mismatch repair-deficient tumors with high microsatellite instability (MSI-H), which account for <5% of metastatic CRC cases[3,4]. Therefore, it is necessary to understand the mechanism of cellular and molecular remodeling in the tumor microenvironment (TME) of CRC and find potential intervention targets to enhance the therapeutic efficacy of immunotherapy. Recent research has revealed that stromal cells and myeloid cells may form a distinctive niche for tumor growth and metastasis[5,6], making them potential therapeutic targets.

Mesenchymal stromal cells represent non-epithelial, non-hematopoietic cell components essential for tissue remodeling, inflammatory response, epithelial cell growth, and immunosuppression[7]. While lacking typical lineage markers, they are positive for vimentin, collagens, PDGFRα/β, and podoplanin with diverse distribution patterns across tissues and cell types[7,8]. Recent advances in single-cell transcriptomics have enabled the systemic profiling of cell populations at an unprecedented degree of resolution in colorectal diseases, including inflammatory bowel disease[9–12], and CRC[13–16]. Diverse stromal cell populations may play distinctive roles in IBD development. For instance, a stromal cell population was reportedly located in the crypt niche with a normal repair and regeneration response function, while another stromal population possessed pro-inflammatory features that contribute to disease severity[9]. In addition, inflammatory fibroblasts expressing IL13RA2 and IL11 were associated with resistance to anti-TNF treatment in IBD patients[10]. The heterogeneity of stromal cells is also thought to be associated with the outcome of CRC progression[14,16]. Two groups of fibroblasts, including myofibroblasts and cancer-associated fibroblasts, have been identified and found to be preferentially enriched in CRC tumors[16]. In addition, myofibroblast-related gene signature has also been identified as one of the major characteristics of the CRC consensus molecular subtype 4, which is defined by the enrichment of tumor stroma and TGF-β signaling mediated extracellular matrix remodeling[17]. Emerging evidence suggests that the tumor myofibroblasts could be a tumor-driven stromal population[14]. However, because the stromal cells used from the above studies represented only a small fraction of the total sequenced population, which prevented their in-depth investigation at a high resolution, it is urgently needed to understand the definitive function of stromal subtypes, especially with regards to their crosstalk with other cells in the TME. In this regard, single-cell RNA sequencing (scRNA-seq) has revealed the importance of cell crosstalk within many types of cancer and identified the interactions of endothelial cells with macrophages[18] and tumor-specific keratinocytes with other types of cells within the TME[19].

It has been reported recently that exclusion of infiltrating immune cells from the TME was associated with poor prognosis for CRC patients[20], and tumor-associated macrophages (TAMs) localizing on the tumor margin have been suggested to prevent the infiltration of cytotoxic lymphocytes (CTL) into the tumor core[21]. Two types of TAMs with distinct inflammatory and angiogenic signatures, and opposite responses to the CSF1R blockade treatment were reported[16]. Other studies also reported a positive correlation between macrophages which expressed markers of M2 macrophages (e.g., CD163, DC-SIGN) or cancer-associated fibroblasts which expressed FSP1, FAP, and poor

outcome for CRC patients[22]. A subtype of TAM with a unique feature called SPP1+ macrophages was reported recently to carry immunosuppressive property and positively correlated with EMT markers as a potential target for anti-tumor growth and metastasis[23]. However, the interaction between myeloid cells and other types of cells in the CRC TME has not been sufficiently investigated.

Here, we identify the presence of diversified tumor microenvironment landscape in colorectal cancer, in which FAP+ fibroblasts and SPP1+ macrophages are enriched in the tumor tissue. This work further highlights that the infiltration of FAP+ fibroblasts and SPP1+ macrophages are highly correlated, and their presence is negatively correlated with lymphocyte infiltration and predicted a poor patient survival. This interaction is validated by immunofluorescent staining and spatial transcriptomics approach, and their co-existence is associated with enriched extracellular matrix expression, thus promoting the formation of tumor desmoplastic structure. Furthermore, high expression of either FAP or SPP1 contributes to resistance to PD-L1 blockade immunotherapy. Together, this work unravels the complex interplay between stromal and macrophages subsets, which could serve as potential targets for CRC immunotherapy.

## Results

**A single-cell transcriptomic atlas of paired human normal mucosa and CRC tissues.** To elucidate the cellular composition of colorectal tumors, tumor samples and adjacent normal tissue were surgically obtained from five non-metastatic patients (Supplementary Table 1; colonic adenocarcinoma (COAD), n = 2; rectal adenocarcinoma (READ), n = 3). The specimens were immediately processed for 3'-end single-cell (sc) RNA-seq using the 10× Genomics platform (Fig. 1a). After filtering the scRNA-seq data to exclude damaged or dead cells and putative cell doublets, a total of 54,103 cell transcriptomes from the five patients were retained for subsequent analysis, in which 29,481 cells were originated from adjacent non-malignant tissues and 24,622 cells from tumors (Fig. 1b). Following gene expression normalization for sequencing depth and mitochondrial read count, we applied principal component analysis (PCA) based on highly variably expressed genes across the sequenced cells. To correct the batch effect, we integrated the scRNA-seq data for tumor and adjacent normal tissue with the Harmony algorithm[24]. We further employed the Harmony-corrected principal components to generate a unified UMAP embedding space and then performed graph-based clustering and annotated each cluster with their respective markers (Supplementary Data 1). The cells were classified into nine major cell types (Fig. 1b–d), including epithelial cells (n = 8940) identified by the expression of EPCAM and CDH1[10], T/ILCs cells (n = 17,420) which expressed the T-cell receptor (TCR) signaling mediators CD3E and CD3G[15], B cells (n = 2998) marked by MS4A1 and CD79A[11], plasma cells (n = 7252) identified by SDC1 and MZB1 expression[25], myeloid cells (n = 4617) which were positive for CD14 and FCGR3A expression[26], mast cells (n = 2781) defined by their classical markers KIT, IL1RL1, and MS4A2, endothelial cells (ECs; n = 2205) marked by PECAM1 and CDH5[11], mesenchymal stromal cells (MSCs; n = 7451) marked by COL1A1 and COL3A1[11], and glial cells (n = 439) marked by S100B and CDH2[10]. Although all nine major cell types were presented in both tumor and adjacent normal tissues from the five patients (Fig. 1e), the grade of infiltration for each of these major cell types was different, possibly reflecting differences in the stage of CRC progression.

**Characteristics of cell populations in tumor tissues from CRC patients reveals hallmark signatures of TME and prediction of clinical outcome.** To investigate the changes in the regulatory

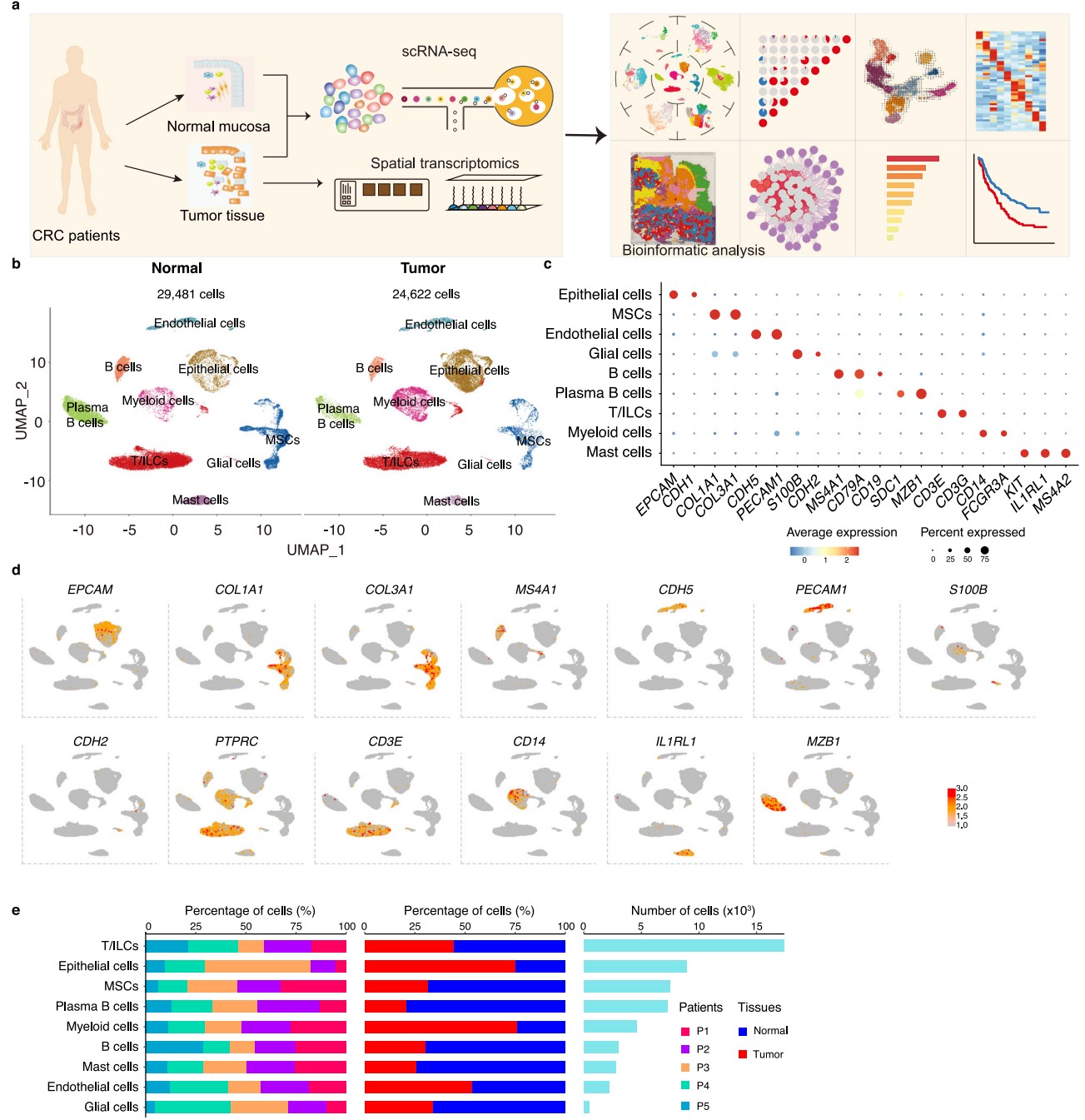

**Fig. 1 Single-cell Atlas of paired human normal mucosa and CRC tissues. a** Graphic overview of this study design. Normal mucosa and tumor tissue from CRC patients were processed into single-cell suspension and unsorted cells were used for scRNA-seq with 10x Genomics. Tumor slides were processed to obtain spatial transcriptomics by 10x Genomics Visium. The following integrated analysis of single-cell transcriptome data is described in squares. **b** UMAP plots of 29,481 cells from normal mucosa and 24,622 cells from tumor tissue of 5 CRC patients, showing 9 clusters in each plot. Each cluster was shown in different color. R package *harmony* was used to correct batch effects and constructed one UMAP based on all cells from adjacent tissue and tumor, and then split cells by these two tissues. **c** Dot plots showing average expression of known markers in indicated cell clusters. The dot size represents percent of cells expressing the genes in each cluster. The expression intensity of markers is shown. **d** Expression levels of selected known marker genes across 54,103 unsorted cells illustrated in UMAP plots from both normal and tumor tissue in CRC patients. **e** Proportion of 9 major cell types showing in bar plots in different donors (left panel), tissues (middle panel), and total cell number of each cell type (right panel) are shown. CRC, colorectal cancer. Source data are provided as a Source data Fig. 1b–e.

networks of tumor-infiltrating cell subsets, we utilized hallmark gene sets of the Molecular Signatures Database (MsigDB)[27] to analyze the alterations in pathways of MSCs, ECs, glial cells, myeloid cells, T cells, and B cells between adjacent normal and tumor tissues (Fig. 2a). The immune-related pathways, including inflammatory response, IL2/STAT5 signaling, and IL6/JAK/ STAT3 signaling, were enriched not only in immune cell populations, such as myeloid cells and T/ILC cells, but also in MSCs and ECs in tumor compared to normal tissues (Fig. 2a), suggesting the involvement of MSCs and ECs in the immune response against colorectal cancer. Hallmark gene set for hypoxia was more enriched in MSCs, ECs, and myeloid cells from tumors

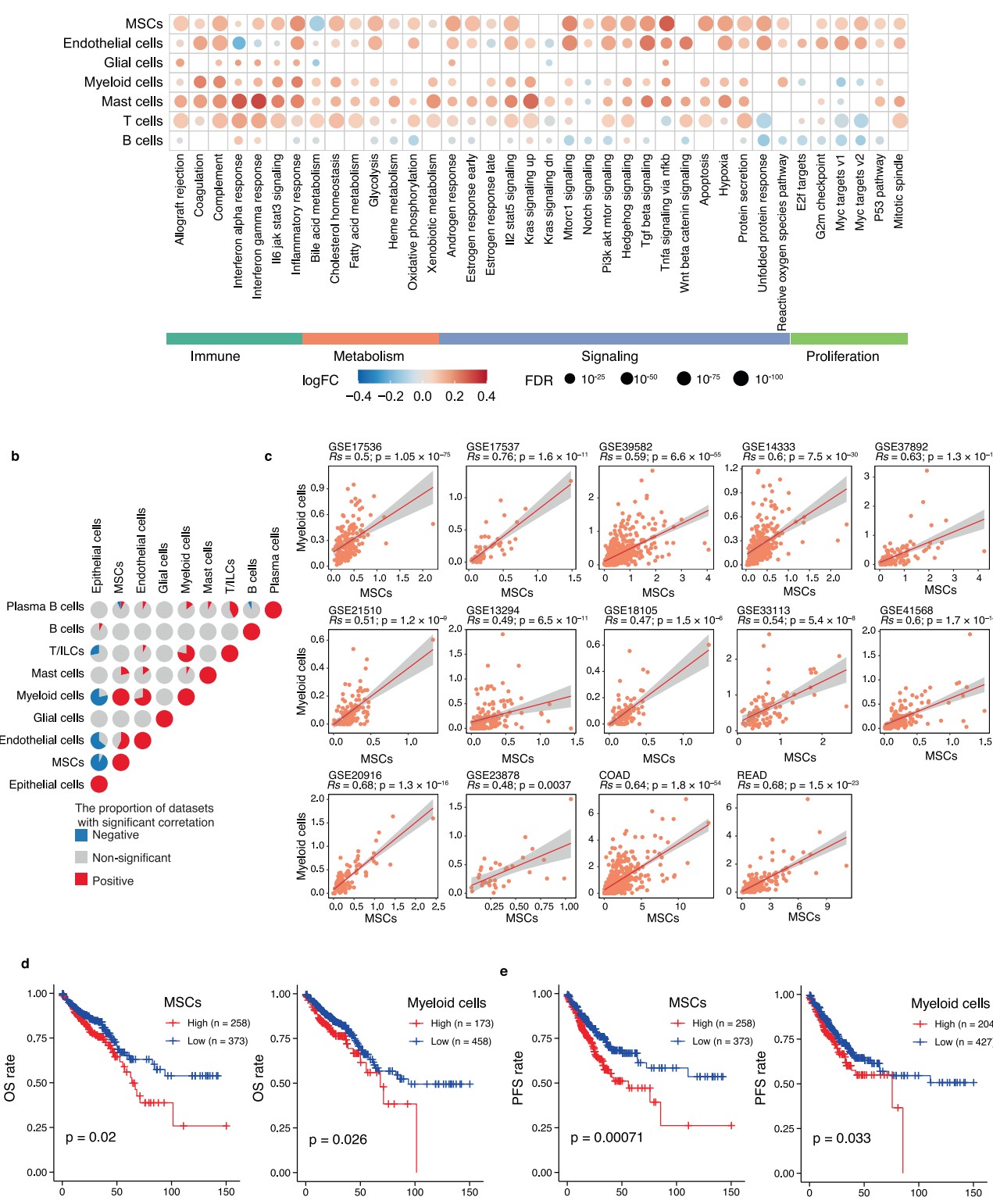

than those of normal samples (Fig. 2a). These characteristics might reflect a stromal cell interaction localized in the hypoxic region of the tumor that links macrophages, MSCs, and ECs to remodel the CRC microenvironment. In addition, tumor-infiltrating myeloid cells and T/ILC exhibited greater

enrichment of metabolism-related genes, including fatty acid metabolism, xenobiotic metabolism, bile acid metabolism, and cholesterol homeostasis, than those cells from normal mucosa (Fig. 2a), suggesting immunometabolism was reprogrammed in the CRC TME. Taken together, these findings indicated that the

**Fig. 2 MSCs and myeloid cells are positively correlated in tumor infiltration and associated with the clinical outcome. a** Dot plots of 50 hallmarks for differentially expressed genes in the global cell type between tumor and adjacent normal tissues. The intensity represents average fold change of gene expression in tumor versus normal mucosa. Dot size shows FDR for each hallmark. Wilcoxon signed-rank test was used to assess the difference. **b** The proportion of CRC cohorts with positive (Spearman correlation; correlation coefficient [Rs] >0.3 and FDR < 0.05, in red), negative (Rs < −0.3 and FDR < 0.05, in blue), or non-significant (gray) correlation for the infiltration of pairwise cell types in 14 independent CRC cohorts. **c** Scatter plots show the correlation between the infiltration of MSCs and myeloid cells across 14 independent datasets with CRC, including TCGA COAD/READ ($n = 635$); GSE39582 ($n = 566$); GSE14333 ($n = 290$); GSE17536 ($n = 177$); GSE13294 ($n = 155$); GSE41568 ($n = 133$); GSE41568 ($n = 133$); GSE37892 ($n = 130$); GSE21510 ($n = 123$); GSE20916 ($n = 111$); GSE18105 ($n = 94$); GSE33113 ($n = 90$); GSE17537 ($n = 55$); GSE23878 ($n = 35$). The error band indicates 95% confidence interval. **d, e** The Kaplan–Meier curves showed patients with higher infiltration of MSCs (left) or myeloid cells (right) are associated with worse OS (**d**) and PFS (**e**) in TCGA COAD/READ cohort. FDR, false discovery rate. MSCs, mesenchymal stromal cells; OS, overall survival, PFS, progressive free survival; TCGA, The Cancer Genomic Atlas; COAD, colon adenocarcinoma; READ, rectum adenocarcinoma. A paired two-sided Wilcoxon signed-rank test was used to assess the difference in (**a**). A two-sided log-rank test p < 0.05 is considered as a statistically significant difference in (**d**, **e**). Source data are provided as a Source data Fig. 2a–e.

regulatory pathways of major cell types were shaped in the CRC TME.

As the sample size of our scRNA-seq dataset was limited, we utilized a deconvolution algorithm CIBERSORTx[28] to simulate the cell-type-specific gene expression profiles to predict the abundance of each cell types quantified by scRNA-seq in large scale datasets from the COAD and READ cohort of The Cancer Genome Atlas (TCGA) and 12 independent CRC cohorts from the Gene Expression Omnibus (GEO; Supplementary Table 2)[29–40]. The robustness of CIBERSORTx to predict cell-type-specific gene expression profiles from TCGA and GEO datasets was trained using our own scRNA-seq dataset (Supplementary Note 1; Supplementary Fig. 1). To determine the relationship among the different cell populations in the CRC microenvironment, we analyzed pairwise Spearman correlations within the infiltration patterns of nine major cell types across 14 independent CRC cohorts. We observed a significantly positive correlation between MSCs and myeloid cells in all interrogated cohorts (Spearman correlation coefficient [|Rs|] > 0.3 and false discovery rate [FDR] < 0.05 were considered significant correlation), with the Rs ranging from 0.47 in the GSE18105 dataset with 98 CRC samples to 0.76 in the GSE17537 dataset with 55 CRC samples (Fig. 2b, c). To assess the clinical relevance for the infiltration of each cell type in CRC TME, we examined the correlation of cell-type infiltration with the overall survival (OS) and progression-free survival (PFS) of CRC patients. These analyses revealed that CRC patients in the TCGA CRC cohort (Fig. 2d, e) with higher MSC infiltration were associated with worse OS (log-rank test, $p = 0.02$) and PFS (log-rank test, $p = 0.00071$)[17], which is consistent with the previously reported results of mesenchymal-type CRC. In addition, greater infiltration of myeloid cells was also associated with worse OS (log-rank test, $p = 0.026$) and PFS (log-rank test, $p = 0.033$; Fig. 2d, e). Furthermore, we revealed a positive correlation between MSCs infiltration and the infiltration of myeloid cells in TME (Fig. 2b, c), and both MSCs and myeloid cells were enriched in CMS4 type of CRC (Supplementary Fig. 2i, j).

**Tumor-specific FAP+ fibroblasts are associated with colorectal cancer progression.** The differences in infiltrated cell types between tumor-adjacent tissues and tumor tissues suggest that a dynamic remodeling of TME plays an important role in CRC progression (Supplementary Note 2). MSCs and fibroblast-like cells have long been suggested as a key stromal cell type involved in regulating tumorigenesis and the progression of cancer[41–45]. However, the identity of these heterogeneous cell population remains elusive. We employed specific cellular signature markers[9] reported before to cluster MSCs into 10 subtypes (Fig. 3a). Telocytes were positive for *SOX6*, *FOXL1*, and *F3* expression[9], which were further sub-clustered based on *ICAM1* expression into *ICAM1+* telocytes ($n = 1016$) and *ICAM1−* telocytes ($n = 550$)

(Fig. 3a, Supplementary Fig. 2c). Myofibroblasts were clustered based on the high expression of contractile genes, *ACTG2* and *MYH11*[9], and further classified into *DES+* myofibroblasts and *MFAP5+* myofibroblasts (Fig. 3a, Supplementary Fig. 2c). *CD24+* fibroblasts ($n = 228$) were characterized by the expression of *CD24* and Wnt agonist gene, *RSPO3*, capable of supporting intestinal stem cell niche[46] (Fig. 3a, Supplementary Fig. 2c). The other fibroblast subtypes included *NT5E+* (encoding CD73, essential for adenosine production and immune suppression) fibroblasts[47] ($n = 930$), *FGFR2+* fibroblasts[48] ($n = 2023$), and *FAP+* (canonical CAF activation marker) fibroblasts[49] ($n = 1091$) (Fig. 3a, Supplementary Fig. 2c). In addition, the proliferating fibroblasts were marked by the expression of *MKI67* (Fig. 3a, Supplementary Fig. 2c).

We next performed data integration between publicly available single-cell transcriptomics data of CRC MSCs and our own datasets to validate tumor-specific *FAP+* fibroblasts (Supplementary Note 3). We compared the differential infiltration of MSC subtypes between tumor and adjacent normal tissue in each donor (Supplementary Fig. 4a). The *FAP+* fibroblasts (Diff = 42.4%, $p = 0.0052$), proliferating fibroblasts (Diff = 2.29%, $p = 0.0099$), and pericytes (Diff = 10.4%, $p = 0.031$) were markedly enriched in tumor tissue as compared to that from adjacent normal tissue, while *NT5E+* fibroblasts (Diff = 14.7%, $p = 0.0053$), *FGFR2+* fibroblasts (Diff = 19.3%, $p = 0.015$), *ICAM1−* telocytes (Diff = −6.53%, $p = 0.0071$), and *MFAP5+* myofibroblasts (Diff = 9.84%, $p = 0.0066$) were enriched in tumor-adjacent normal tissue (Fig. 3b, c, Supplementary Figs. 2d and 4b). There is no significant difference of *CD24+* fibroblasts, *DES+* fibroblasts, and *ICAM1+* telocytes between the tumor and tumor-adjacent normal tissues (Fig. 3b, Supplementary Figs. 2d and 4b). To further validate the changes in MSCs subtypes, we estimated the abundances of cell types in large datasets from TCGA CRC and the 12 other independent CRC cohorts described above. The analysis was performed using CIBERSORTx with cell-type-specific gene expression profiles for 58 cell types defined by our scRNA-seq. In agreement with the results discussed above, *FAP+* fibroblasts were significantly enriched in the CRC samples ($p = 3.7 \times 10^{-8}$; Fig. 3d). *FAP+* fibroblasts expressed the markers *FAP*, *MMP1*, and *MMP3* (Supplementary Data 2) typically associated with fibroblast activation and extracellular matrix remodeling. Interestingly, CRC patients with higher degrees of *FAP+* fibroblasts infiltration exhibited shorter PFS in both the TCGA and dataset GSE17536 related cohorts (Fig. 3e, Supplementary Fig. 4c). In addition, the infiltration of *FAP+* fibroblasts was promoted at the late cancer stage and was higher in patients with MSI-H in the TCGA CRC cohort (Supplementary Fig. 4d). To validate this tumor-specific fibroblast infiltration, we analyzed the expression of stromal cell subtype markers CD24, CD26, NT5E, FAP, and FGFR2 in colorectal tumor and non-malignant large intestine samples by flow cytometry (Fig. 3f, g). The results showed

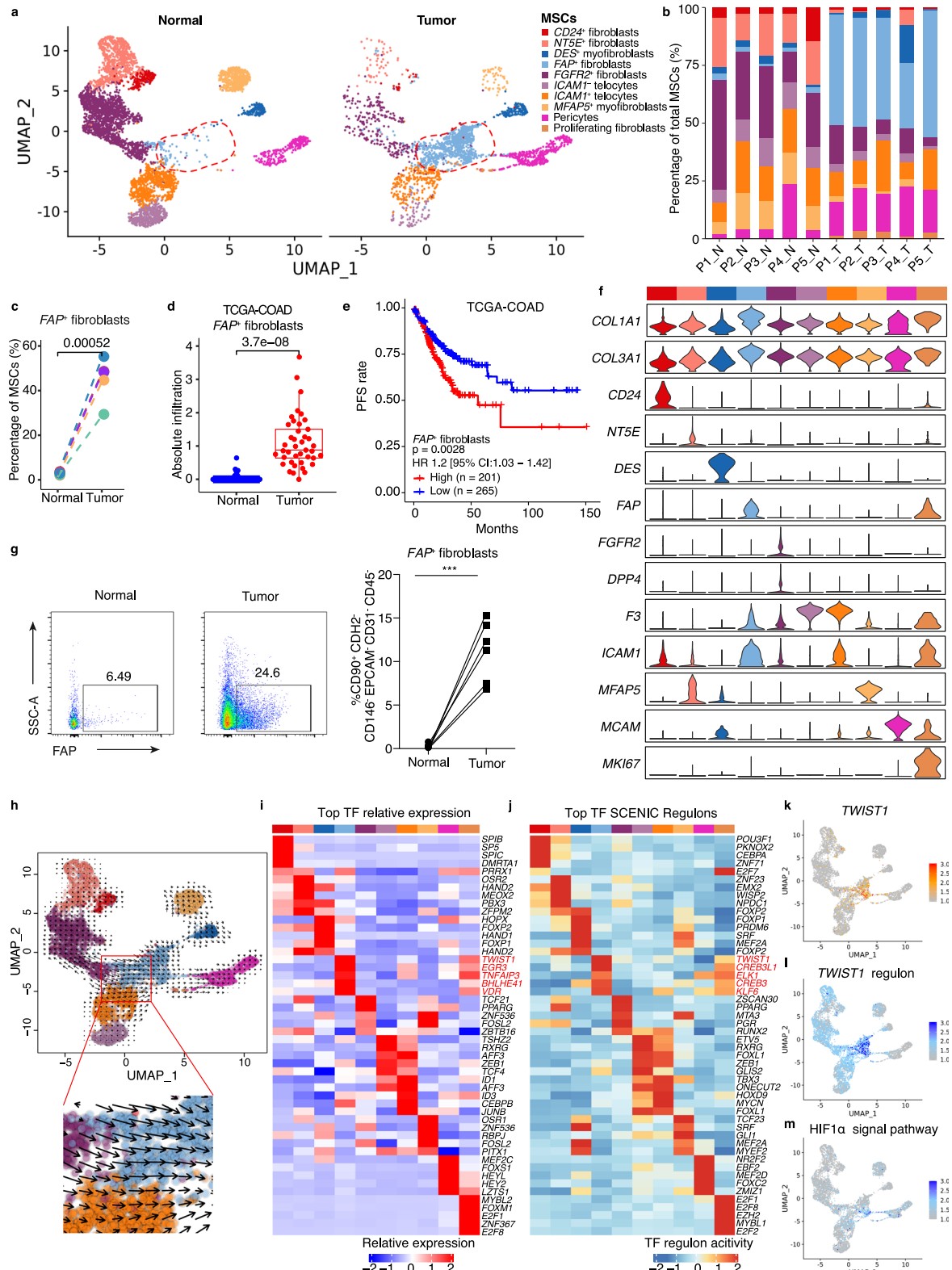

the infiltration of *FAP⁺* fibroblasts were significantly increased, while *NT5E⁺* and *FGFR2⁺* fibroblasts were significantly decreased in CRC samples (Fig. 3g, Supplementary Fig. 4e, f).

Then, we performed RNA "velocity" analysis, a computational approach that utilizes nascent transcription in scRNA-seq datasets to infer differentiation trajectories of *FAP⁺* fibroblasts[50]. This analysis predicted that tumor-specific *FAP⁺* fibroblasts likely originated from *FGFR2⁺* fibroblasts or *ICAM1⁺* telocytes

(Fig. 3h). The differentiation of *FAP⁺* fibroblasts is orchestrated by a sophisticated network of transcription factors (TFs) that regulate each other and their effectors by interacting with their cofactors and downstream genes. Therefore, we evaluated the top five specifically expressed TFs and the top five activities of the TF regulatory network by pySCENIC[51]. We found that *TWIST1* had the highest expression and activity levels in the regulatory network of *FAP⁺* fibroblasts and may represent the master TF

**Fig. 3 Characterization of stromal cells in normal mucosa and tumor tissue. a** UMAP showing the composition of MSCs colored by cluster. Red dashed circle shows $FAP^+$ fibroblasts. **b** Bar plots show the percentage of each MSCs subtypes in scRNA-seq. **c** Comparison of frequencies of $FAP^+$ fibroblasts of MSCs in paired normal mucosa ($n = 5$) and tumor ($n = 5$) tissue in scRNA-seq. **d** Comparison of absolute infiltration proportion of $FAP^+$ fibroblasts between paired normal ($n = 41$) and tumor ($n = 41$) in TCGA-COAD cohort. Boxes show the median ± 1 quartile, with the whiskers extending from the hinge to the smallest or largest value within 1.5× the IQR from the box boundaries. **e** The Kaplan–Meier progression-free survival curves of COAD patients stratified by $FAP^+$ fibroblasts infiltration. **f** Violin plots showing the expression of selected genes. Colors as in (**a**). **g** Representative flow cytometry plots (left) and proportion of $FAP^+$ fibroblasts (right) in normal mucosa ($n = 6$) and tumor ($n = 6$) tissue. Gating strategies of $FAP^+$ fibroblasts were as in Supplementary Fig. 4e. $p = 0.0006$. **h** RNA velocity of MSCs subtypes. Color as in (**a**). Inferred developmental trajectory of $FAP^+$ fibroblasts is red-circled and enlarged (down). **i** Heatmap shows the relative expression (z-score) of top 5 transcription factors (TFs) genes in each MSCs subtypes as in (**a**). **j** Heatmap shows normalized activity of top 5 TF regulons in MSCs subtype predicted by pySCENIC. Clusters are colored as in (**a**). TF regulons with top activity for $FAP^+$ fibroblasts are color-coded in red. **k**–**m** UMAP plots showing expression of $TWIST1$ (**k**), the activity of $TWIST1$-regulon (**l**), and the enrichment score HI1α signal pathway (**m**). Cell is colored by the z-score normalized value. UMAP, Uniform Manifold Approximation and Projection; P, patient. N, normal mucosa; T, tumor tissue; CRC, colorectal cancer. A paired two-sided Student's $t$-test was used to assess the difference in (**c**), (**d**), and (**g**). A two-sided log-rank test was used to assess statistical significance in (**e**). $p < 0.05$ is considered as a statistically significant difference. ***$p < 0.001$. Source data are provided as a Source data Fig. 3a–g, k–m.

driving this differentiation pathway (Fig. 3i–l). Hypoxia is one of the most important characteristics of the TME, and a previous study demonstrated that $TWIST1$ might be regulated by hypoxia[52,53]. Indeed, the hypoxia-dependent HIF-1α signaling pathway was significantly enriched in $FAP^+$ fibroblasts compared with other MSC subtypes (Fig. 3k).

To understand the differentiation of $FAP^+$ fibroblasts from their two potential precursors is regulated, we analyzed the differentially expressed genes and performed KEGG enrichment analysis (Supplementary Fig. 4g–j). In addition to the above-mentioned $TWIST1$, when compared to $FGFR2^+$ fibroblasts and $ICAM1^+$ telocytes, respectively, $FAP^+$ fibroblasts showed higher expression of $PRRX1$ (Supplementary Fig. 4g, h), which is critical for tuning cancer-associated fibroblast activation and plasticity in cancer[54]. In addition, compared with $ICAM1^+$ telocytes, $FAP^+$ fibroblasts showed marked expression of $FAP$, $IGFBP4$ (encodes a growth factor), $FN1$ (encodes a glycoprotein of the extracellular matrix), and $HES4$ (important for transcription factor binding and protein dimerization) (Supplementary Fig. 4h). Furthermore, $FAP^+$ fibroblasts were enriched in ECM-receptor interaction and focal adhesion-related pathways compared with either $FGFR2^+$ fibroblasts or $ICAM1^+$ telocytes (Supplementary Fig. 4i, j), implying the involvement of these cells in the ECM formation. We further demonstrated that genes highly expressed in $FAP^+$ fibroblasts compared with all other MSCs subtypes were mostly the genes involved in ECM-receptor interaction, TNF signaling pathway, and fatty acid biosynthesis, suggesting that activation of these processes is involved in the commitment of $FAP^+$ fibroblasts (Supplementary Fig. 4k).

**Tumor-specific $SPP1^+$ macrophages are associated with CRC progression.** The remodeling of myeloid cells of CRC patients suggests the cells have functional roles in tumorigenesis. Three dendritic cell (DC) subtypes were identified (Fig. 4a). Activated DCs ($n = 73$) were marked by high expression of $CCR7$, required for innate lymphoid cell trafficking to draining lymph nodes[55], and $FSCN1$, essential for DC maturation[56] (Fig. 4a and Supplementary Fig. 2c). $XCR1$ and $CLEC9A$ were highly expressed in cDC1 ($n = 91$), while $FCER1A$ and $CD1C$[57] were upregulated in cDC2 ($n = 487$) (Fig. 4a and Supplementary Fig. 2c). $THBS1^+$ macrophages ($n = 1028$) were characterized by $THBS1$ expression (Fig. 4a and Supplementary Fig. 2c), which was shown to activate M1-like TAMs and promote the malignant migration of cancer[58]. In addition, $VCAN^+$ macrophages ($n = 600$) were identified by high expression of $VCAN$, and another macrophage subtype, $C1QC^+$ $MRC1^-$ macrophages ($n = 1480$), were positive for expression of the complement component gene $C1QC$ but lacked $MRC1$ expression (Fig. 4a and Supplementary Fig. 2c). $SPP1^+$

macrophages showed high expression of $SPP1$ and the scavenger receptor $MARCO$[59] ($n = 443$) (Fig. 4a and Supplementary Fig. 2c). Furthermore, we observed a neutrophil population characterized by $CSF3R$ expression[57] and proliferating myeloid cells known for $MKI67$ expression (Fig. 4a and Supplementary Fig. 2c). Furthermore, we integrated our single-cell transcriptomics of myeloid cells with data from two previous studies to supplement the information on the myeloid landscape in CRC (Supplementary Note 4).

We investigated the alterations in myeloid cell subtypes among adjacent tissues and tumor tissues and showed that macrophage and neutrophil populations were predominantly present in tumor tissues, while DCs were enriched in adjacent normal tissues (Fig. 4a, b and Supplementary Fig. 6a, b). Importantly, $SPP1^+$ macrophages are tumor-specific macrophages, accounting for 11.6% of myeloid cells in tumor samples but only 0.68% of the myeloid cells in adjacent normal tissues (Fig. 4c). $SPP1^+$ macrophages in CRC expressed the $C1QC$, $MRC1$, $STAT1$, and $PPARG$ (Supplementary Data 2) markers typically associated with the polarization of macrophages. Consistent with these results, the infiltration of $SPP1^+$ macrophages in tumor samples were found to be significantly upregulated compared with adjacent normal tissues in the TCGA cohort when imputed cell infiltration by CIBERSORTx. CRC patients in both TCGA and GSE17536 CRC cohorts with a higher infiltration of $SPP1^+$ macrophages exhibited shorter PFS (Fig. 4e, Supplementary Fig. 6c), and the infiltration correlated with late-stage cancer and MSI-H patients in the TCGA CRC cohort (Supplementary Fig. 6d). Based on the flow cytometric analysis of myeloid cell subset markers CD14, CD206 (encoded by $MRC1$), CD209, and CD13 (encoded by $ANPEP$) (Fig. 4f), we further documented that $SPP1^+$ macrophages were significantly increased in CRC compared with non-malignant colon tissue while $THBS1^+$ macrophages were non-significant change (Fig. 4g, Supplementary Fig. 6e, f). Furthermore, RNA velocity predicted that tumor-specific $SPP1^+$ macrophages probably originated from $THBS1^+$ macrophages (Fig. 4h). $SPP1^+$ macrophages showed higher expression of genes encoding calcium- and zinc-binding proteins, including $S100A10$, $S100A8$, and $S100A6$, and genes related to the extracellular matrix, such as $CD44$ (Supplementary Fig. 6g). Moreover, $SPP1^+$ macrophages were potentially regulated by the IL-17 signaling pathway, HIF-1 signaling, and cytokine-cytokine receptor interaction, while $THBS1^+$ macrophages were capable of performing antigen-processing and presentation, and regulating intestinal immune network for IgA production (Supplementary Fig. 6h), implying that $SPP1^+$ macrophages and $THBS1^+$ macrophages execute distinct functions in TME.

To further determine the master regulator of $SPP1^+$ macrophages, we performed pySCENIC analysis. Results indicated that

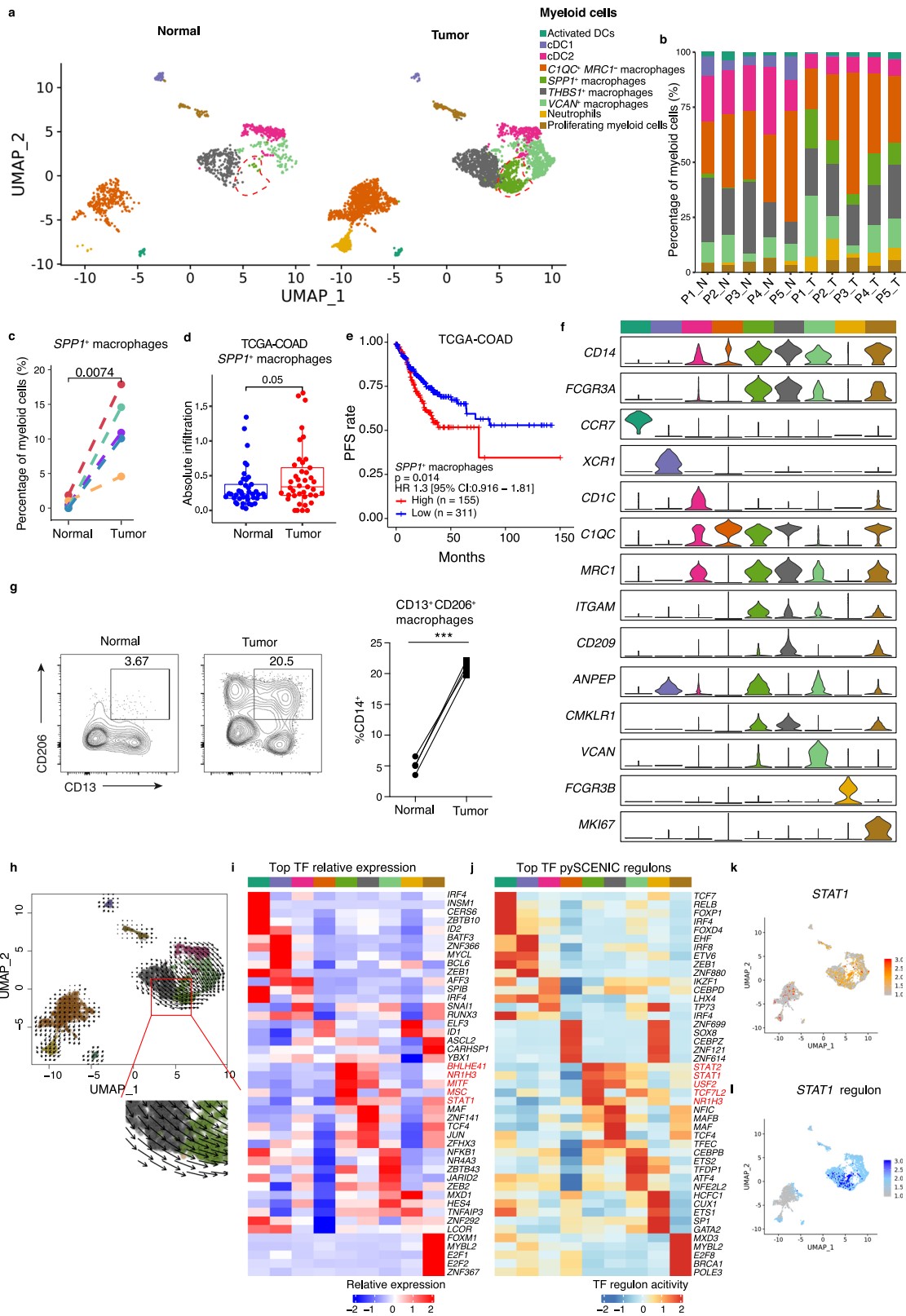

STAT1, encoding master transcription factor that skews macrophages toward an M1 phenotype in the mouse, was highly active in SPP1+ macrophages[60] (Fig. 4i–l). Genes highly expressed in SPP1+ macrophages were those involved in ECM−receptor interactions, the PPAR signaling pathway, and glycolysis/gluconeogenesis (Supplementary Fig. 6i). Interestingly, SPP1+ macrophages were also potentially regulated by the HIF-1

signaling pathway (Supplementary Fig. 6h, i), which is the typical hypoxia-induced pathway. These findings suggest that activation of these signaling pathways is involved in the commitment of SPP1+ macrophages. Because FAP+ fibroblasts and SPP1+ macrophages were closely involved in the hypoxia-induced pathway, we hypothesized the presence of a stromal cell-mediated network localized in the hypoxic region of the tumor

**Fig. 4 Characterization of myeloid cells in normal mucosa and tumor tissue. a** UMAP of individual myeloid cells. Red dashed circle shows *SPP1+* macrophages. Each dot denotes one cell; color represents cluster origin. **b** Bar plots showing proportion of each myeloid subtype in each sample. **c** Comparison of *SPP1+* macrophage percentages in paired normal mucosa (*n* = 5) and tumor (*n* = 5) tissue. **d** Comparison of absolute infiltration proportion of *SPP1+* macrophage between paired normal (*n* = 41) and tumor (*n* = 41) in TCGA-COAD cohort. The boxes show the median ± 1 quartile, with the whiskers extending from the hinge to the smallest or largest value within 1.5× the IQR from the box boundaries. **e** The Kaplan–Meier curve shows COAD patients survival with different *SPP1+* macrophages infiltration. **f** Violin plots showing expression of selected genes. Color of each cluster refers to (**a**). **g** Representative flow cytometry plots (left) and frequencies of *SPP1+* macrophages (right) in normal mucosa (*n* = 4) and tumor (*n* = 4) tissue of CRC patients. Gating strategies are shown in Supplementary Fig. 6e. *p* = 0.0002. **h** Grid visualization of RNA velocity for myeloid cell subtypes on a UMAP embedding. Cells are colored as in (**a**). Inferred developmental trajectory of *SPP1+* macrophage is red-circled and enlarged (down). **i** Heatmap shows relative expression (z-score) of the top 5 highly expressed transcription factors (TFs) in each cell subtypes. Clusters were colored as in (**a**). **j** Heatmap normalized activity of top 5 TF regulons in MSCs subtype predicted by pySCENIC. Clusters are colored as in (**a**). **k, l** UMAP plots show expression of *STAT1* in myeloid cells (**k**), the activity of STAT1-regulon (**l**). Cell is colored by the z-score normalized value. UMAP, Uniform Manifold Approximation and Projection; P, patient. N, normal mucosa; T, tumor tissue; CRC, colorectal cancer. A paired two-sided Student's *t*-test was used to assess the difference in (**c**), (**d**), and (**g**). A two-sided log-rank test was used to assess statistical significance in (**e**). *p* < 0.05 is considered as a statistically significant difference. ***p* < 0.001. Source data are provided as a Source data Fig. 4a–g, k–i.

that links macrophages and MSCs, which collaborate to exacerbate the CRC microenvironment.

**High infiltration of *FAP+* fibroblasts and *SPP1+* macrophages correlates with worse patient survival.** *FAP+* fibroblasts and *SPP1+* macrophages were mostly enriched in tumor tissue (Figs. 3b and 4b), and a high correlation between the infiltration of MSCs and myeloid cells was found in patients across 14 colorectal cancer datasets (Fig. 2b). To investigate the infiltration status of these subsets, we then used CIBERSORTx to assess the infiltration of 58 cell clusters identified by scRNA-seq in 14 independent CRC cohorts and further calculated the pairwise Spearman correlations within the infiltrations of these cell clusters in each cohort (Fig. 5a). We found that the *FAP+* fibroblasts and *SPP1+* macrophages were the most highly correlated populations across all examined cohorts (Fig. 5a, Supplementary Fig. 7). To further uncover the clinical implication of such close correlation between these two cell types, we compared the PFS of patients with different levels of *FAP+* fibroblasts and *SPP1+* macrophages. Patients with both high *FAP+* fibroblasts and *SPP1+* macrophages exhibited the shortest PFS compared with the signature combination groups, suggesting that these two cell types can synergistically promote tumor progression (Fig. 5b).

To further understand the potential triggers or downstream signals that induce *FAP+* fibroblasts and *SPP1+* macrophages, we calculated the differentially expressed genes between *FAP+* fibroblasts^High *SPP1+* macrophages^High and *FAP+* fibroblasts^Low *SPP1+* macrophages^Low groups in a TCGA-COAD cohort and performed GSEA analysis subsequently. Genes upregulated in samples with *FAP+* fibroblasts^High *SPP1+* macrophages^High showed enrichment of epithelial–mesenchymal transition signatures (Fig. 5c). These samples also displayed highly enriched hypoxia gene set (Fig. 5c), which is consistent with the supposed hypoxic environment of tumors[53]. Furthermore, TNFα signaling and IL2/STAT5 signaling pathways were also enriched in *FAP+* fibroblasts^High *SPP1+* macrophages^High tumor samples (Fig. 5c). These findings suggest that *FAP+* fibroblasts and *SPP1+* macrophages respond to different stimuli and signals within the TME of CRC. Then, we assessed the protein expression levels by the H-score system in tissue microarray (TMA) of 78 CRC patients to identify FAP and SPP1 both specifically increased in tumor tissue compared with adjacent normal tissue (Fig. 5d, e) and found patients with high protein levels of FAP or SPP1 enrolled in this TMA cohort exhibited shorter OS (Fig. 5f, g). Furthermore, patients with high protein levels of both FAP and SPP1 survived poorly compared to patients with lower level of FAP or SPP1 (Fig. 5h). In addition, none of the patients in TMA dataset expressed high levels of SPP1 but low FAP, probably due

to our relatively small sample size of only 78 individuals. We further examined whether *FAP+* fibroblasts and *SPP1+* macrophages localize closely in CRC tissues. Immunofluorescent labeling demonstrated the close proximity of SPP1-positive and FAP-positive cells in CRC tissue (Fig. 5I, j), implying there is potential crosstalk between these two cells.

**Cell–cell interaction of *FAP+* fibroblasts and *SPP1+* macrophages revealed by spatial transcriptomics.** To further assess the spatial organization of *FAP+* fibroblasts and *SPP1+* macrophages, we performed spatial transcriptomics (ST) with tumor tissue sections from four CRC patients (Fig. 6a, d, Supplementary Fig. 9a, d). Transcriptomics from 4457, 4248, 3890, and 1657 spots were obtained at a median depth of 7518, 6618, 4830, and 16,868 UMIs/spot, 3083, 2778, 2051, and 4937 genes/spot, and 10.66%, 10.63%, 14.37%, and 11.35% mitochondrial genes/spot for patient #6, #7, #8, and #9, respectively (Supplementary Fig. 8). Based on the unbiased clustering and spot features, we classified the spots into 10 clusters, *FAP+* fibroblasts/*SPP1+* macrophages, malignant epithelial cells, epithelial cells, MSCs, *MT+* cells with fibrosis, *FAP+* fibroblasts, myofibroblasts, endothelial cells, immune cells, and unknown of patient #6 (Fig. 6a–c). We also classified the spots of CRC patient #7 and #8 into six clusters, i.e., *FAP+* fibroblasts/*SPP1+* macrophages, malignant epithelial cells, MSC cells, myofibroblast cells, immune cells, and unknown (Fig. 6d–f, and Supplementary Fig. 9a–c), classified the spots of CRC patient #9 into 8 clusters, *FAP+* fibroblasts/*SPP1+* macrophages, malignant epithelial cells, epithelial cells, MSCs, endothelial cells, myofibroblasts, immune cells, and unknown (Supplementary Fig. 9d–f). Score spots in each cluster with *FAP+* fibroblasts or *SPP1+* macrophage signatures from scRNA-seq data (top 25 specifically expressed genes) highlighted a cluster with *FAP+* fibroblasts and *SPP1+* macrophages co-localization in the same spot (Fig. 6g, h and Supplementary Fig. 9g–m) and wrapped around malignant epithelial cells (Fig. 6i). In addition, the signature score of *FAP+* fibroblasts or *SPP1+* macrophages showed significantly positive correlation (Fig. 6j, k and Supplementary Fig. 9n–o). Most *FAP+* fibroblasts and *SPP1+* macrophages were colocalized at the same spots, and a spot in the 10x Genomics Visium platform could accommodate up to 10 cells, suggesting that there was a physical interaction between the two cell types. Furthermore, we found spots with *FAP+* fibroblasts and *SPP1+* macrophages highly active in pathways that contribute to the formation of desmoplastic structures in four ST datasets, including extracellular matrix organization, collagen fibril organization, and response to TGF-β (Fig. 6l). T cells or B cells were excluded from tumor core (Fig. 6m, Supplementary Fig. 9p–r). These results suggest the desmoplastic

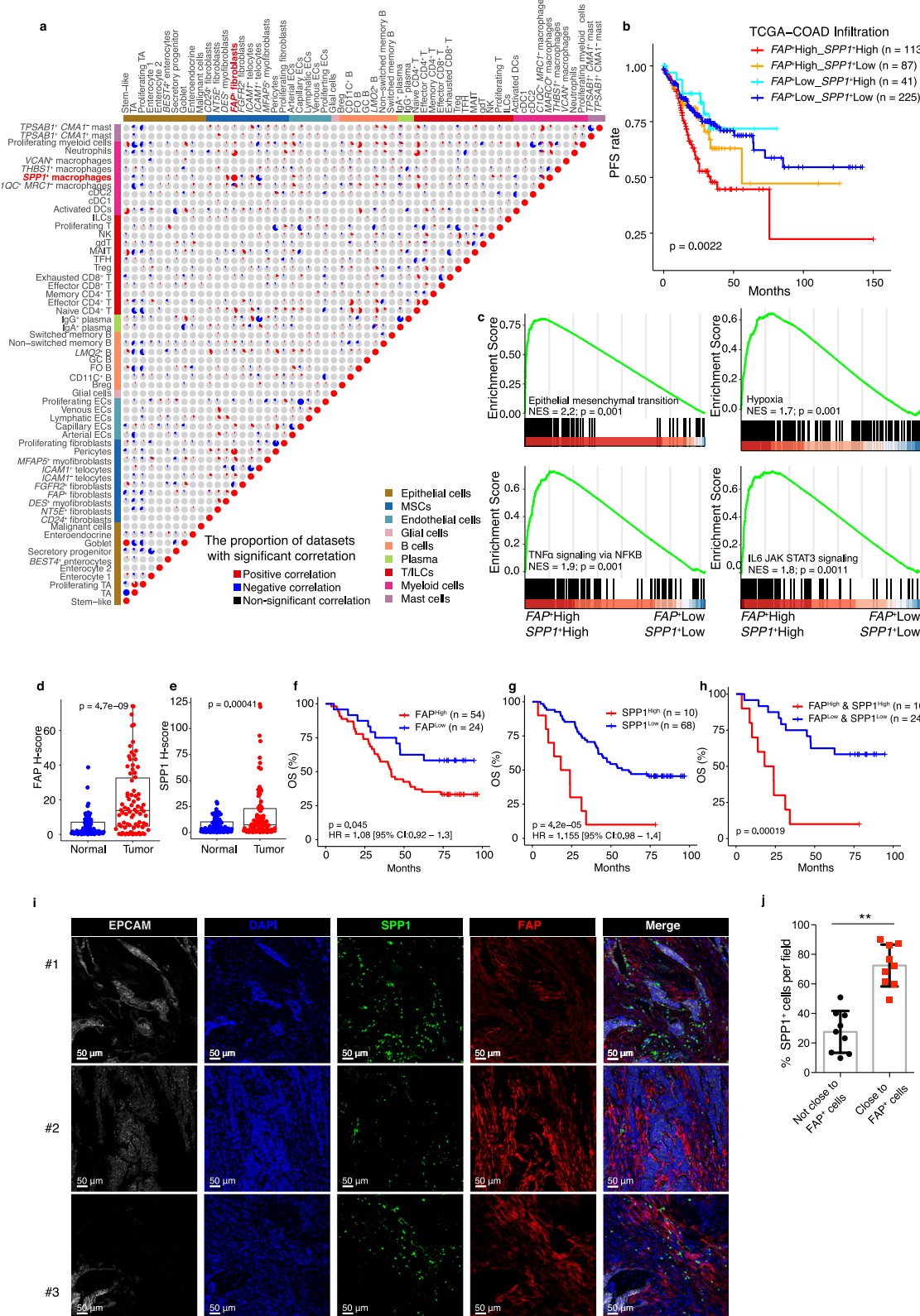

microenvironment may be regulated by the interaction of *FAP*+ fibroblasts and *SPP1*+ macrophages to limit the infiltration of immune cells into the tumor core.

**FAP+ fibroblasts and SPP1+ macrophages interaction may contribute to desmoplastic tumor microenvironment.** To

further identify the key mediators of *FAP*+ fibroblasts and *SPP1*+ macrophages interaction in CRC patients, we investigated the cell–cell communication mechanisms of *FAP*+ fibroblasts and *SPP1*+ macrophages. To this end, we evaluated the putative crosstalk with the R package "NicheNet" based on the expression and downstream targets of ligand-receptor pairs[61]. We found that *FAP*+ fibroblasts could directly contact with *SPP1*+ macrophages

**Fig. 5 High infiltration of *FAP*⁺ fibroblasts and *SPP1*⁺ macrophages associated with worse patient survival and immunotherapy resistance. a** Pie charts show the proportion of CRC cohorts with positive (Spearman correlation; correlation coefficient [Rs] > 0.3 and FDR < 0.05, in red), negative (Rs < −0.3 and FDR < 0.05, in blue), or non-significant (gray) correlation for the infiltration of pairwise 58 cell types in 14 independent CRC cohorts. **b** Progressive free survival analyses for four subgroups of TCGA-COAD patients stratified by the infiltration of both *FAP*⁺ fibroblasts and *SPP1*⁺ macrophages using Kaplan–Meier curves. **c** GSEA of epithelial–mesenchymal transition, hypoxia, TNFα signaling via NFKB, and IL2 STAT5 signaling pathways between *FAP*⁺ fibroblasts high/*SPP1*⁺ macrophages high and *FAP*⁺ fibroblasts low/*SPP1*⁺ macrophages low groups. Ranking genes by fold change in expression between these two conditions. NES, normalized enrichment score. **d, e** Quantification of FAP (**d**) and SPP1 (**e**) staining intensity in the adjacent normal (*n* = 78) and CRC tissue (*n* = 78) microarray. **f–h** Kaplan–Meier curves showed overall survival analyses for low and high expression FAP alone (**f**), SPP1 alone (**g**), and both (**h**) CRC patients. **i** Representative IF staining of human CRC tissue (20x). EPCAM (gray), DAPI (blue), SPP1 (green), FAP (red), in individual and merged channels are shown. Bar, 50 μm. Experiment was performed in three independent patients. **j** Proportion of *FAP*⁺ fibroblasts colocalized with SPP1-positive cells (*n* = 9, 3 patients with 3 sections). Data represent mean ± SD. *p* = 0.0014. A two-sided log-rank test was used to assess statistical significance in (**b**) and (**f–h**). A paired two-sided Student's *t*-test was used to assess the difference in (**d**), (**e**), and (**i**). Source data are provided as a Source data Fig. 5a–i.

through the adhesive ligand-receptor pairs *COL1A1/LAMA1-ITGB1* (Fig. 7a). In addition, *FAP*⁺ fibroblasts enhanced the pro-inflammatory activity of *SPP1*⁺ macrophages via the expression of TGF-β superfamily genes, *TGFB1* and *INHBA*, and TGF-β induced expression of *ACVRL1*, *ACVR1*, or *ACVR1B* in *SPP1*⁺ macrophages (Fig. 7a). Furthermore, *FAP*⁺ fibroblasts enhanced the recruitment of *SPP1*⁺ macrophages through the *WNT5A-FZD2* and *CCL3-CCR5* pairs. Of note, *FAP*⁺ fibroblasts interacted with *SPP1*⁺ macrophages through *RARRES2-CMKLR1* (Fig. 7a). We further determined the relative expression of *RARRES2* across MSC clusters and found this gene to be expressed at a higher level in *FAP*⁺ fibroblasts than other MSC clusters (Fig. 7b). Chemerin, encoded by *RARRES2*, was shown to be an independent risk factor for CRC and has the ability to affect macrophage polarization in the DSS-induced colitis model[62,63]. In addition, *CMKLR1*, encoding the receptor for chemerin, showed higher expression in both *THBS1*⁺ macrophages and *SPP1*⁺ macrophages (Fig. 7c). Since RNA "velocity" analysis indicated that *SPP1*⁺ macrophages can be differentiated from *THBS1*⁺ macrophages (Fig. 4e), *FAP*⁺ fibroblasts might function as a driver, promoting *SPP1*⁺ macrophage differentiation through chemerin. Furthermore, we found that chemerin levels were significantly higher in the plasma of CRC patients than in healthy donors (Fig. 7d), suggesting that chemerin may serve as a predictive marker for CRC.

Fibroblasts are the main producers of extracellular components, including extracellular matrix components, which may contribute to the formation of desmoplastic structures[64]. Genes of extracellular matrix (ECM)-related pathways were highly expressed in *FAP*⁺ fibroblasts and *SPP1*⁺ macrophages, suggesting that either *FAP*⁺ fibroblasts or *SPP1*⁺ macrophages may facilitate the generation of desmoplastic structures (Supplementary Figs. 4f and 6g). Therefore, we investigated whether *SPP1*⁺ macrophages promote the ECM remodeling ability of *FAP*⁺ fibroblasts. As revealed by NicheNet analysis, *SPP1*⁺ macrophages showed high *TGFB1*, *IL1B*, and *IL1A* ligand activity and relatively high *TGFB1*, *IL1B*, and *IL1A* gene expression (Fig. 7e, f). In addition, *TGFB1* encoding protein bound to receptors encoded by *TGFBR3*, *ACVRL1*, and *TGFBR1* on *FAP*⁺ fibroblasts, whereas ligands encoded by *IL1B* or *IL1A* interacted with receptors encoded by *IL1R1* or *IL1RAP* on *FAP*⁺ fibroblasts (Fig. 7g), resulting in the expression of target genes encoding collagen or matrix metallopeptidase in these cells (Fig. 7h). These targets are important components of desmoplastic reactions, and 35 of the 100 predicted targets encoded components of ECM pathways[65], including extracellular niche fibrous components (e.g., collagens [encoded by *COL10A1*, *COL11A1*, *COL1A1*, *COL1A2*, *COL3A1*, *COL5A1*, *COL8A1*], fibronectin [encoded by *FN1*], and integrins [encoded by *ITGA5*, *ITGB5*]), remodeling proteins (e.g., the lysyl oxidase family [encoded by *LOX*, *LOXL1*,

*LOXL2*]), and matrix metalloproteinases (encoded by *ADAM17*, *MMP1*, *MMP14*, *MMP2*, *MMP3*, *TIMP1*, *TIMP2*, *TIMP3*) (Fig. 7h). We further performed NicheNet ligand activity analysis of the gene set of KEGG pathways and the extracellular matrix. Indeed, there was a high probability that the target genes belonged to cytokine-cytokine receptor interaction, extracellular matrix pathways, the TNF signaling pathway, and the TGF-beta signaling pathway (Fig. 7i). We further explored the signaling pathways among the top three ligands (encoded by *TGFB1*, *IL1A*, and *IL1B*) and ECM-related targets, and we found 40 downstream regulators, including *HIF1A*, *AKT1*, *STAT1*, and *NFKB1*, those are involved in signaling pathways connecting ligands secreted by *SPP1*⁺ macrophages and ECM-target genes that contribute to the formation of the desmoplastic region (Fig. 7j). Taken together, our findings indicate that *FAP*⁺ fibroblasts and *SPP1*⁺ macrophages form an interaction network supporting each other's maintenance and function. These two cell types may play important roles in the remodeling of ECM, potentially promoting the formation of the desmoplastic region of the TME.

**High infiltration of *FAP*⁺ fibroblasts and *SPP1*⁺ macrophages correlates with immunotherapy resistance.** The tumor immune microenvironment can generally be classified into immune-excluded, inflamed (referred to as "hot" tumors, which respond to immunotherapy well), and immune-desert types based on the infiltration of immune cells and CD8⁺ T cells[5,66]. We were, therefore, curious about the local immune features of tumors with different infiltration patterns of *FAP*⁺ fibroblasts and/or *SPP1*⁺ macrophages. Interestingly, CRC tumors samples in the *FAP*⁺ fibroblasts^High *SPP1*⁺ macrophages^High group showed a relatively high rate of non-silent mutations and single-nucleotide variant (SNV)-predicted neoantigens. In addition, they displayed a higher Shannon entropy index and TCR richness, reflecting a non-effective T-cell response to neoantigens (Fig. 8a–d). This subtype of CRC also displayed the lowest lymphocyte infiltration among all CRC subtypes (Fig. 8e), suggesting that the micro-environmental characteristics of this certain type of tumors is immune-exclusive with high infiltration of both *FAP*⁺ fibroblasts and *SPP1*⁺ macrophages. Current tumor immunotherapy mainly targets lymphocytes, therefore, the reduced lymphocyte infiltration of this type of tumor would dampen the effect of immunotherapy. To test this hypothesis, we performed association analysis of survival time and *FAP* or *SPP1* expression in patients with bladder cancer treated with anti-PD-L1 using IMvigor210 dataset[67]. Patients with high levels of either *FAP* or *SPP1* expression showed impaired OS in response to PD-L1 antibody treatment (Fig. 8f, g). Furthermore, anti-PD-L1-treated patients with high levels of both *FAP* and *SPP1* expression showed shorter survival times (Fig. 8h). Importantly, patients expressing high levels of either *FAP* or *SPP1* showed a lower response (complete response [CR] and partial response [PR]), but more progressive disease (PD) than patients with

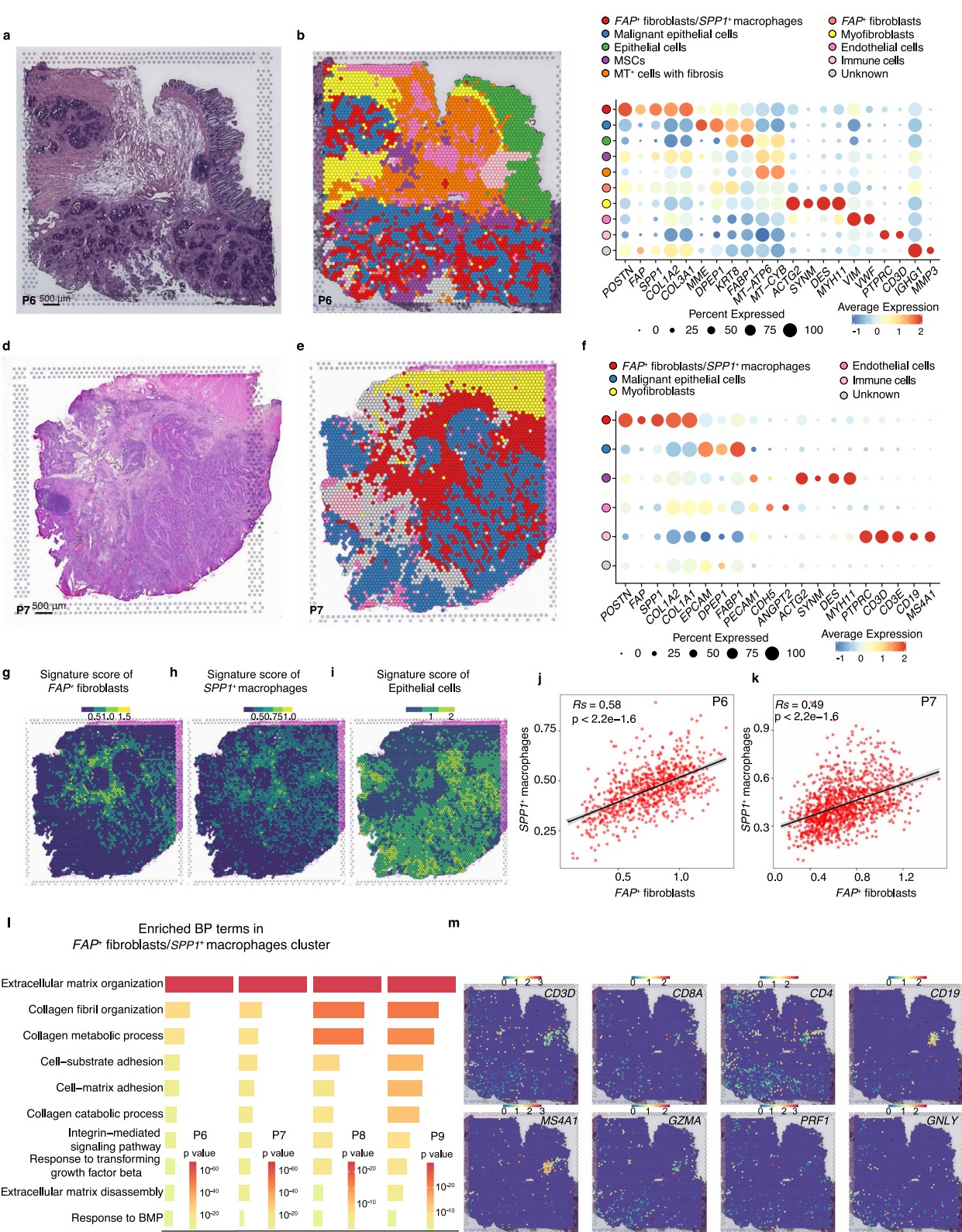

low expression of both *FAP* and *SPP1* (Fig. 8i), suggesting that *FAP*- or *SPP1*-enriched patients have a significantly poorer response to anti-PD-L1 treatment than other patients.

## Discussion

Although cancer immunotherapy has recently achieved remarkable success, it was not effective for everyone, and the mechanism of non-responsiveness is not fully understood. In clinical trials of CRC patients received ICB therapies, there has been relatively limited information obtained on predictive biomarkers and immunotherapy strategies[1]. Previous single-cell transcriptomic studies of CRC focused either on certain cell types[13,15,16,68,69] or on CRC classification based on consensus molecular subtypes[14]. There is an urgent need to apply the single-cell transcriptomic

**Fig. 6 Co-localization of *FAP*[+] fibroblasts and *SPP1*[+] macrophages revealed by spatial transcriptomics. a** Hematoxylin and eosin (H&E) staining of tissue sections of ST spots in CRC patient #6. Scale bar, 500 μm. Experiment was repeated twice for one tumor section. **b** Unbiased clustering of ST spots and define cell types of each cluster. **c** Dot plots showing average expression of known markers in indicated cell clusters. The dot size represents percent of cells expressing the genes in each cluster. The expression intensity of markers is shown. **d–f** H&E staining of tissue sections (**d**), unbiased clustering (**e**) of ST spots, and dot plots showing average expression of known markers in indicated cell clusters (**f**) in CRC patients #7 as (**a–c**) shown. H&E staining was repeated twice for one tumor block. **g–i** Spatial feature plots of signature score of *FAP*[+] fibroblasts (**g**), *SPP1*[+] macrophages (**h**), and epithelial cells (**i**) in tissue sections. **j, k** The Pearson correlation of signature score of *FAP*[+] fibroblasts (x axis) and *SPP1*[+] macrophages (y axis) in *FAP*[+] fibroblasts/*SPP1*[+] macrophages cluster in patient #6 (**j**) and patient #7 (**k**). The error band indicates 95% confidence interval. **l** Gene ontology (GO) terms of genes significantly enriched in *FAP*[+] fibroblasts/*SPP1*[+] macrophages cluster in four patients. The statistical analysis was performed by Fisher's test. **m** Spatial feature plots of gene expression of *CD3D, CD8A, CD4, CD19, MS4A1, GZMA, PRF1,* and *GNLY* in tissue sections of patient #6. Source data are provided as a Source data Fig. 6b, c, e–m.

analysis to larger CRC cohorts and the analysis of the interactions between two tumor-specific clusters or their impacts on immunotherapy. Through integrated analyses of scRNA-seq performed in this study, publicly accessible scRNA-seq and bulk RNA-seq datasets, spatial transcriptomics, FACS, IF, and transcriptomics with ICB treatment, our study provided a comprehensive picture of the landscape of TME and its adjacent normal tissue counterpart at the single-cell resolution, as well as demonstrating that the interaction between *FAP*[+] fibroblasts and *SPP1*[+] macrophages may play an instructive role in the formation of desmoplastic TME, which may lead to resistance to tumor immunotherapy.

The heterogeneity of fibroblasts played important roles in modulating the tumor immune microenvironment[70]. Recent single-cell transcriptomic analyses have revealed, to some extent, the heterogeneity of stromal cells. These analyses included the investigation of *ACTA2*[+] myofibroblasts and cancer-associated fibroblasts expressing *FAP*[16] and the characterization of the myofibroblasts and stromal 1/2/3 clusters[14]. These studies revealed that cancer-associated fibroblasts correspond to activated fibroblasts, and they typically express markers, such as FAP, FSP, and αSMA, and play an important function in tumor modulation and chemotherapy resistance[71]. In this study, we systemically deciphered the heterogeneity and features of stromal cells and demonstrated the dramatic remodeling of stromal compartments upon tumorigenesis. The remodeling involves a particular increase in a cancer-associated fibroblast subtype, i.e., *FAP*[+] fibroblasts, and a decrease in a mesenchymal stem-like cell subtype, i.e., *NT5E*[+] fibroblasts. Further trajectory analysis indicated that *FAP*[+] fibroblasts might be generated by the TME-controlled differentiation of *FGFR2*[+] fibroblasts or *ICAM1*[+] telocytes, and *TWIST* plays a key regulatory role in this commitment. *FAP*[+] fibroblasts are actively involved in tissue remodeling by affecting ECM-receptor interactions or microenvironment metabolism involving non-essential amino acids, galactose, and steroid or fatty acid biosynthesis. Our results on the heterogeneity of stroma cell subtypes might explain, at least in part, the findings of previous contradictory studies, in which the use of different deletion systems to analyze IKKβ yielded contradicting effects, either enhancing tumor growth or decreasing inflammation and suppressing tumor growth[72–74]. Thus, our study has highlighted the importance of further research on the function and tracing system of different stromal subtypes.

Due to its dipeptidyl peptidase and collagenase activity, FAP participates in tissue repair, fibrosis, and extracellular matrix degradation[71,75]. Depletion of FAP-expressing cells in transgenic mice carrying Lewis lung carcinoma or subcutaneous pancreatic ductal adenocarcinoma resulted in rapid hypoxic necrosis of both cancer and stromal cells, and the process was mediated by TNFα and IFNγ[76]. However, phase II clinical trials of a single agent *Talabostat* or humanized monoclonal antibodies targeting FAP have failed in patients with metastatic CRC[77,78]. This lack of

efficacy might be a result of neglecting the complex cell interaction networks in the tumor environment. This study used multiple independent CRC cohorts with large sample sizes to perform association analysis of different cell subtypes, and we found that myeloid cell subtype *SPP1*[+] macrophages are the most relevant cells that interact with MSC subtype *FAP*[+] fibroblasts and are associated with shorter OS and PFS of CRC patients.

Further analysis of myeloid cell subtypes revealed a dramatic increase in *SPP1*[+] macrophages in tumor tissue. One of the most significant features of *SPP1*[+] macrophages was the expression of *SPP1*, implying that this subtype may correspond to the *SPP1*[+] macrophages described by Zhang et al.[16]. We identified that the *C1QC*[+] *MRC1*[−] macrophage subtype in colon tissue was closely related to CD14/CD16 monocytes identified in the blood by Zhang et al., possibly reflecting the loss of this population during tissue preparation[16]. In agreement with other studies on macrophages, we did not detect candidate human counterparts of mouse M1 and M2 macrophages[16,79,80]. As validated by FACS, the *SPP1*[+] macrophages expressed the typical M2 marker CD206, but showed high M1 macrophage regulator activity of *STAT1* by pySCENIC analysis[81–83]. Higher infiltration of *SPP1*[+] macrophages correlated with shorter PFS in CRC patients. More effort is needed to investigate the key transcription factors that govern the *SPP1*[+] macrophages and determine their function.

The TME can be categorized into three different types, including the inflamed type, in which immune cells infiltrate but are inhibited, the immune-excluded type, and the immune-desert type[5]. In our study, we dissected the potential mechanism of the generation of the immune-excluded microenvironment of CRC. We investigated the interactions between *FAP*[+] fibroblasts and *SPP1*[+] macrophages from multiple aspects, including single-cell and spatial transcriptomics, immunofluorescent labeling, and imputation analysis of other datasets. We predicted that *FAP*[+] fibroblasts might promote the differentiation of *THBS1*[+] macrophages into *SPP1*[+] macrophages through the *RARRES2/CMKLR1* interaction. ECM remodeling is necessary for generating desmoplastic regions, and *SPP1*[+] macrophages promote the expression of ECM-related genes in *FAP*[+] fibroblasts by secreting cytokines encoded by *IL1A, IL1B,* or *TGFB1*, suggesting the desmoplastic microenvironment is controlled by *SPP1*[+] macrophages and *FAP*[+] fibroblast interactions. Desmoplastic TME promotes tumor growth, and invasion through ECM remodeling, in which stromal cells play a key role[84,85]. Interestingly, we found that although tumor tissues with high infiltration of both *FAP*[+] fibroblasts and *SPP1*[+] macrophages showed high neoantigen features, they were characterized by non-silent richness and SNV neoantigens. The Shannon index and richness of TCR were relatively high, indicating a diversified T-cell repertoire. Combined with the observation of relatively decreased lymphocyte infiltration and normal lymphocyte infiltration, we inferred that this might represent an immune-excluded TME. In addition, we found that high expression of *FAP* or/and *SPP1* can predict

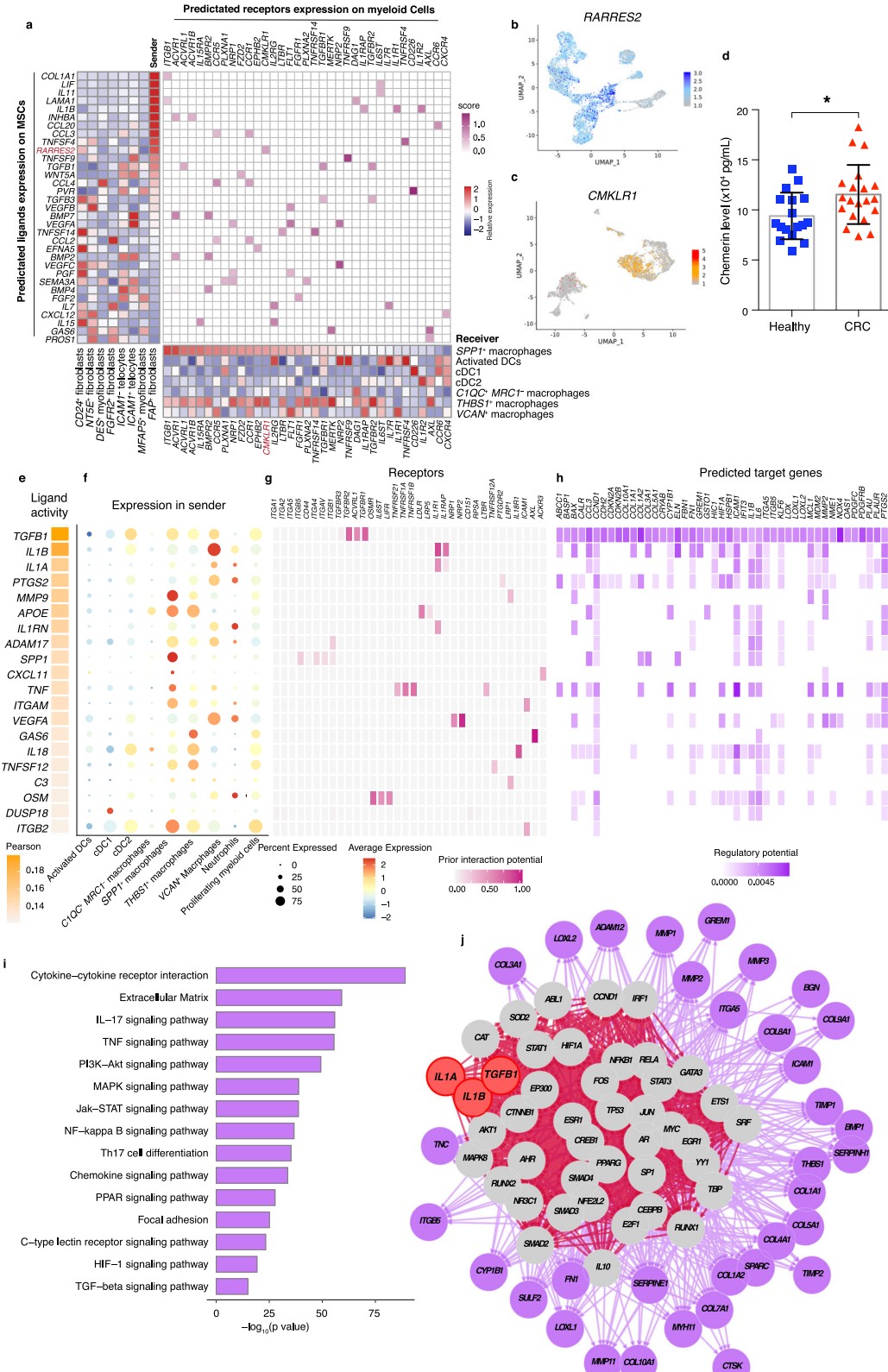

shortening of the OS and a reduced response to anti-PD-L1 therapy. These results confirmed our conclusion of an immune-excluded microenvironment, reflecting a desmoplastic TME in CRC tumors. Furthermore, depletion of *FAP*+ stromal cells in mouse tumor models disrupts the desmoplastic structure and promotes tumor growth[85]. Our work further showed the potential mechanisms involved and strengthened the hypothesis that the interaction between *FAP*+ fibroblasts and *SPP1*+ macrophages should be considered as a target for CRC treatment.

Together, these findings imply that *FAP*+ fibroblasts and *SPP1*+ macrophages contribute to ECM remodeling and coordinate to form a desmoplastic microenvironment that prevents lymphocytes infiltrating the tumor core, further reducing the efficacy of PD-L1 treatment (Fig. 8j). We propose that, mechanistically, *FAP*+

**Fig. 7 The interaction network between _FAP_+ fibroblasts and _SPP1_+ macrophages. a** Heatmap of top predicated ligands expression across MSCs subtypes of NicheNet which was used to predicate ligands expressed by _FAP_+ fibroblasts that modulate _SPP1_+ macrophages (left). Bottom, heatmap of relative receptors expression on different myeloid subtypes. Middle, heatmap of significant ligand-receptor pairs between _FAP_+ fibroblasts and _SPP1_+ macrophages. _FGFR2_+ fibroblasts or _ICAM1_+ fibroblasts as reference cell types for ligand-receptor analysis. **b**, **c** UMAP plots showing the relative expression of _RARRES2_ in MSCs subtypes (**b**) and _CMKLR1_ in myeloid cell subtypes (**c**). Intensity is shown with z-score normalized expression. **d** Statistical analysis comparing the chemerin concentration in plasma of healthy donor ($n = 17$, blue) and CRC patients ($n = 20$, red). The error bars are presented as the mean ± SEM. **e** Top-ranked ligands inferred to regulate _FAP_+ fibroblasts by _SPP1_+ macrophages according to NicheNet. **f** Dot plots showing the expression percentage (dot size) and intensity (dot intensity) of top-ranked ligands (**e**) in each myeloid subtype. **g** Ligand-receptor pairs showing interaction between _SPP1_+ macrophages and _FAP_+ fibroblasts ordered by ligand activity (**e**). **h** Heatmap showing regulatory potential of top 20 ranked ligand (**e**) and the downstream target genes in _FAP_+ fibroblasts. **i** Representative KEGG pathways enrichment of the predicted target genes expressed in _FAP_+ fibroblasts. The statistical analysis was performed by Fisher's test. **j** Diagrams showing the network of top 3 active ligands (encoded by _TGFB1_, _IL1A_, and _IL1B_) for target genes belonging to ECM and potential signaling pathways. An unpaired two-sided Student's _t_-test was performed in (**d**), *$p < 0.5$. UMAP, uniform manifold approximation and projection; MSCs, mesenchymal stromal cells; CRC, colorectal cancer. Source data are provided as Source data Fig. 7b–i and Source data Fig. 7j.rds.

fibroblasts are regulated by _TWIST1_, the expression of which is induced by hypoxia; they also secrete chemerin, which can enter the blood vessels and bind to _THBS1_+ macrophages, which may differentiate into _SPP1_+ macrophages. In addition, _SPP1_+ macrophages might regulate _FAP_+ fibroblasts via _TGFB1_, thus promoting the secretion of MMPs and collagen, which, together, contribute to the remodeling of ECM (Fig. 8j). The biological interaction mechanisms still need to be addressed in future work. Our study has unveiled the detailed landscape of both immune and nonimmune cells in the microenvironment of CRC and highlighted the potential value of identifying and developing therapeutic strategies targeting _FAP_+ fibroblasts, _SPP1_+ macrophages, or the molecules involved in their crosstalk to overcome immune suppression and increase the response of tumors to immunotherapies.

## Methods

**Collection of clinical human samples**. Adjacent normal mucosa and tumor tissues were collected from CRC patients with informed written consent, and under approval of local medical ethnics from Ruijin Hospital Affiliated to Shanghai Jiao Tong University, Renji Hospital Affiliated to Shanghai Jiao Tong University, and The First Affiliated Hospital of University of Science and Technology of China. Fresh tissues were kept in RPMI 1640 containing 10% FBS on ice, and ready for transport.

**Tissue dissociation**. Fat tissue and visible blood vessels were removed before tissue process. Fresh normal mucosa and CRC tissue were washed with ice-cold PBS, and cut into small pieces. For normal mucosa, tissues were placed and shaken into 10 mL of EDTA-containing buffer (5 mM EDTA, 15 mM HEPES, 1 mM DTT, and 10% FBS-supplemented PBS) for 1 h at 37 °C. Tumor tissues were incubated with 10 mL of DTT (65 mM)-containing PBS (supplemented with 10% FBS) for 15 min at 37 °C with shaking. EDTA and DTT were removed after the above incubation with PBS twice. Small tissue pieces were minced and digested with collagenase VIII at 0.38 mg/mL and DNase I at 0.1 mg/mL in complete RPMI 1640 medium (containing 10% FBS, 100 U/mL penicillin, and 100 mg/mL streptomycin) for 1 h at 37 °C. After digestion, tubes were shaken vigorously for 5 min. 21-gauge syringes were used to dissociate cells mechanically. Cells were filtered through 100 μm filter and pelleted and washed with PBS twice. Freshly prepared cell suspensions were ready for scRNA-seq and flow cytometry staining.

**Single-cell RNA sequencing**. Freshly prepared cell suspensions were performed immediately according to the manufacturer's protocol of 10 X Chromium 3' v3 kit (10x Genomics, Pleasanton, CA). Library was prepared and sequencing was performed on a NovaSeq 6000 platform (Illumina, Inc., San Diego, CA) in GENERGY BIO (Shanghai, China).

**Raw 10× read alignment, quality control, and normalization**. Raw sequencing reads were transformed into fastq file with Illumina bcl2fastq2 Conversion Software v2.20 at https://support.illumina.com/downloads/bcl2fastq-conversion-software-v2-20.html, and quality checked with FastQC software v0.11.9, at https://www.bioinformatics.babraham.ac.uk/projects/fastqc/. Standard pipelines of cell ranger were used to do sequence processing, alignment to GRch38 genome with default parameters (https://support.10xgenomics.com/single-cell-gene-expression/software/pipelines/latest/).

**Dimension reduction and clustering analysis**. We scaled data with top 2000 most variable genes by using _FindVariableFeatures_ function in R package Seurat v3. Clustering[86]. We used variable genes for principal component analysis (PCA), used _FindNeighbors_ in Seurat to get nearest neighbors for graph clustering based on PCs, and used _FindCluster_ in Seurat to obtain cell subtypes, and visualized cells with the uniform manifold approximation and projection (UMAP) algorithm. To eliminate the batch effect, we performed harmony algorithm in Harmony R package[24] to remove batch correction before clustering analysis, and applied _FindNeighbors_ and _FindCluster_ in Seurat to obtain cell subtypes. Cells were clustered at two stages of the analysis, partitioned cells into epithelial, stromal, myeloid, T, and B cells in the first stage, then clustered cells from multiple samples into distinct subtypes in the second stage. For the first step, the clusters were scored for the previously described gene signatures[10], including epithelial cells (_EPCAM_, _KRT8_, _KRT18_), stromal cells (_COL1A1_, _COL1A2_, _COL6A1_, _COL6A2_, _VWF_, _PLVAP_, _CDH5_, _S100B_), myeloid cells (_CD68_, _XCR1_, _CLEC9A_, _CLEC10A_, _CD1C_, _S100A8_, _S100A9_, _TPSAB1_, and _OSM_), T cells (_NKG7_, _KLRC1_, _CCR7_, _FOXP3_, _CTLA4_, _CD8B_, _CXCR6_, and _CD3D_), and B cells (_MZB1_, _IGHA1_, _SELL_, _CD19_, and _AICDA_). Signature scores were calculated as the mean $\log_2$(LogNormalizedUMI+1) across all genes in the signature. Each cluster was assigned to the compartment of its maximal score and all cluster assignments were manually checked to ensure the accurate partition of cells. For the second step, we performed harmony algorithm before clustering analysis to remove batch correction, and applied _FindNeighbors_ and _FindCluster_ in Seurat to obtain cell subtypes. As an auxiliary tool, we defined 58 cell types in CRC based on the gene signatures of each cell type and known lineage markers.

**Assessment of the robustness of subclusters in a machine-learning protocol**. We performed the analysis by random forest in R package randomForest, using 50% of each subcluster for training and 50% for testing. Receiver operating characteristic (ROC) curves of multi-class classifications were constructed, using the false-positive rate (x axis) and the true positive rate (y axis) for all possible thresholds of probabilities given by the random forest. The area under the curve (AUC) was then calculated to assess the quality of cluster assignment and considered as the robustness of subclusters identification.

**Differential-expression analysis**. We used the "FindAllMarkers" function in Seurat to identify genes that are differentially expressed between clusters with the following parameters: min.pct = 0.1, logfc.threshold = 0.25, pseudocount.use = 0.1, only.pos = T. The non-parametric Wilcoxon rank-sum test was used to obtain _p_-values for comparisons, and the adjusted _p_-values, based on Bonferroni correction, for all genes in the dataset. We used heatmap to visualize DEGs based on gene expression after the log-transformed and scaling.

**Trajectory reconstruction based on RNA velocity estimation**. To investigate the origin of differentiation for _FAP_+ fibroblasts and _SPP1_+ macrophage, we analyzed expression dynamics by estimating the RNA velocities of single cells by distinguishing between unspliced and spliced transcripts based on loom files of scRNA-seq data. We used the R package velocyto.R (https://github.com/velocyto-team/velocyto.R) to calculate the RNA velocity value of each gene in each cell, and embed the RNA velocity vector in a low-dimensional space, and then visualized it on the UMAP projection using Gaussian smoothing on a regular grid[50].

**Transcription factor regulon analysis**. The analysis of the regulatory network and regulon activity was performed by pySCENIC[51]. The regulon activity (measured in AUC) was analyzed by AUCell module of the pySCENIC, and the active regulons were determined by AUCell default threshold. The differential-expression regulon was identified by Wilcoxon rank-sum test in "FindAllMarkers" function in R package Seurat with following parameters: min.pct = 0.1, logfc.threshold = 0.25,

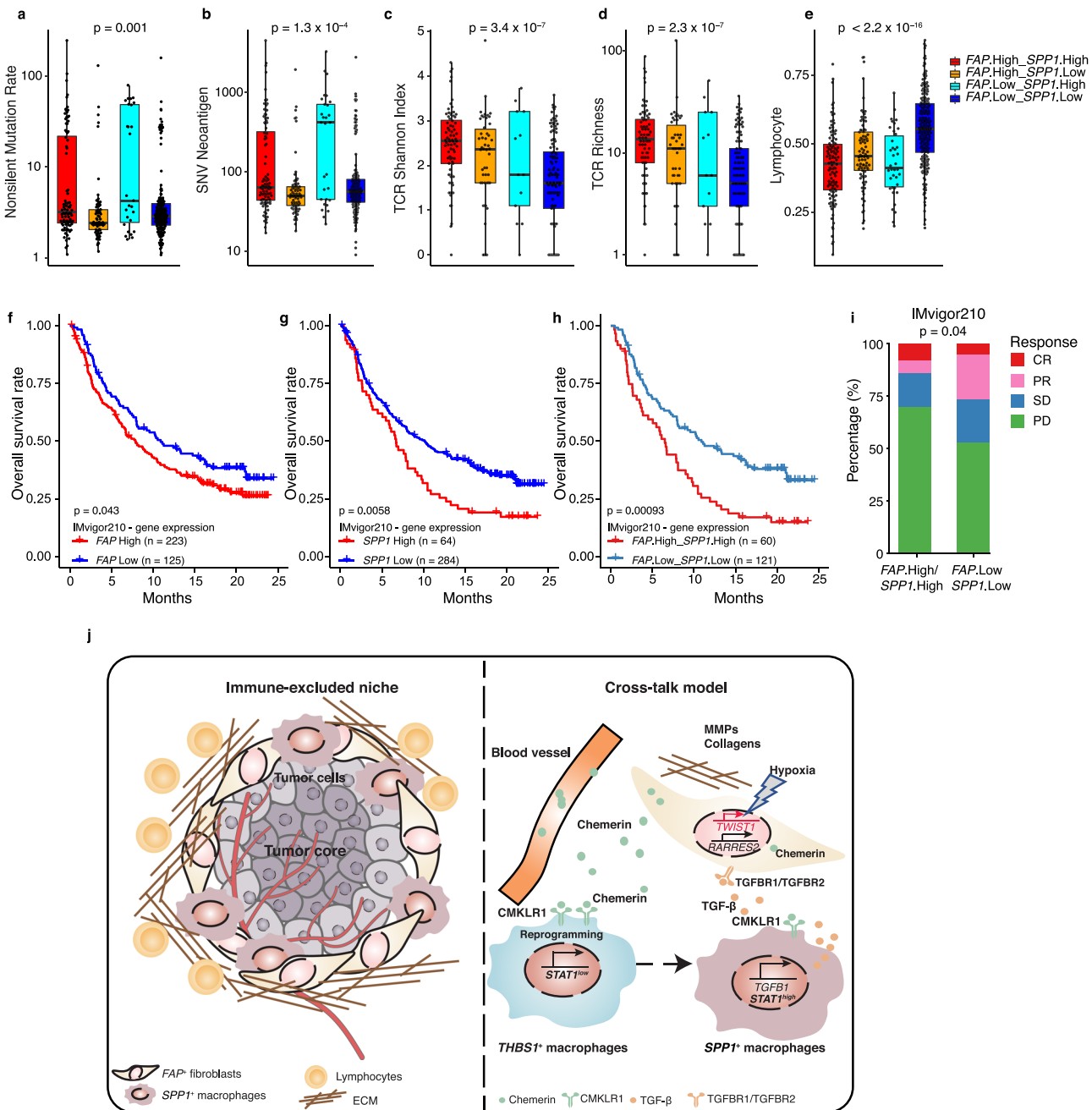

**Fig. 8 The infiltration of lymphocytes in TME and response to PD-L1 blockade in patients affected by *SPP1*+ macrophages and *FAP*+ fibroblasts.**
**a–i** The non-silent mutation rate (**a**), SNV neoantigen (**b**), TCR Shannon index (**c**), TCR richness (**d**), and the infiltration of lymphocyte (**e**) for four subgroups of TCGA-COAD patients stratified by the infiltration of both *FAP*+ fibroblasts and *SPP1*+ macrophages, including *FAP*+High_*SPP1*+High ($n = 113$), *FAP*+High_*SPP1*+Low ($n = 87$), *FAP*+Low_*SPP1*+High ($n = 41$), and *FAP*+Low_*SPP1*+Low ($n = 225$) The boxes show the median ± 1 quartile, with the whiskers extending from the hinge to the smallest or largest value within 1.5× the IQR from the box boundaries. **f–h** Survival analyses for low and high expression of *FAP* alone (**f**), *SPP1* alone (**g**), and both (**h**) patient groups in the anti-PD-L1 immunotherapy cohort using Kaplan–Meier curves (IMvigor210 cohort). **i** Bar plots show patients with both low expression of *FAP* and *SPP1* are more likely to respond to anti-PD-L1 treatment. **j** Working model for the immune-excluded niche and crosstalk model between *FAP*+ fibroblasts and *SPP1*+ macrophages. CRC, colorectal cancer; TCGA, The Cancer Genomic Atlas; COAD, colon adenocarcinoma; GSEA, gene set enrichment analysis; IF, immunofluorescence; TCR, T-cell receptor; SNV, single-nucleotide variant; IQR, interquartile range; CR, complete response; PR, partial response; SD, stable disease; PD, progressive disease. One-way ANOVA test was used in (**a**–**e**). A two-sided log-rank test was used to assess statistical significance in (**f**–**h**). Fisher's test was used in (**i**). Source data are provided as a Source data Fig. 8a–i.

pseudocount.use = F, only.pos = T. The scaled expression of regulon activity was used to generate a heatmap.

To check the gene expression of transcription factors (TFs) alone, we retrieved Genes encoding TFs from four TF-related public datasets: JASPAR[87] (http://jaspar.genereg.net/), DBD[88] (http://www.transcriptionfactor.org/), AnimalTFDB[89] (http://bioinfo.life.hust.edu.cn/AnimalTFDB/), and TF2DNA[90] (http://

www.fiserlab.org/tf2dna_db/). We overlapped the TF genes with the DEGs quantified above, and determined the most specifically expressed TFs in each cluster.

**Characterization of cell-type infiltration based on single-cell expression matrix.** To establish the proportions of our defined 9 major cell types and

58 subdivided cell types from bulk RNA-seq and microarray data, we used the online tool CIBERSORTx[28] to create a reference signature matrix from our single-cell RNA-seq dataset and estimate cell-type proportions from 14 public bulk RNA-seq and microarray datasets based constructed cell-type reference. Our 9 major cell types and 58 subdivided cell types reference dataset consisted of 54103 cells, each of which has been annotated as the major or subdivided cell types as we described earlier. For creating signature matrices, CIBERSORTx was run with quartile normalization was disabled for RNA-seq datasets and was enabled for microarray datasets, and all other parameters with their default settings. For the imputation of cell fractions, the quartile normalization was disabled for RNA-seq datasets and was enabled for microarray datasets, the permutation parameter was set to 500 times, and all other parameters are kept at their default settings. Spearman's correlation analysis was performed to assess the relationship among the proportions of cell-type infiltration, and considered | $Rs$ | > 0.3 and FDR < 0.05 as significant correlation. The unsupervised clustering of R package *Heatmap.plus* was used to assess and visualize the Euclidean distance of cell infiltration among all cell types in CRC cohorts.

**Survival analysis**. Survival analysis was performed by R package *survival*. Hazard ratio (HR) was calculated by Cox proportional hazards model and 95% CI was reported, and Kaplan–Meier survival curve was modeled by survfit function. The "maxstat.test" function of R package *maxstat* which all potential cutting points were repeatedly tested to find the maximum rank statistic, was used to perform dichotomy of cell population infiltration or gene expression, and then divide the patients into two groups according to the selected maximum logarithm statistics. The two-sided long-rank test was used to compare Kaplan–Meier survival curves. The comparison of the percentage of patients who respond to ICB treatment between different groups was determined by Chi-squared test.

**Cancer hallmarks and GSEA analyses**. Fifty hallmark gene sets were downloaded from The Molecular Signatures Database[27] (MSigDB, http://software.broadinstitute.org/gsea/msigdb/). Scoring these gene signatures was as previously described[10] (https://www.github.com/cssmillie/ulcerative_colitis). In briefly, the gene signature score for each cell was calculated by the mean scaled expression across all genes in the signatures. To identify statistically significant alteration of gene signature between normal and tumor samples, we used "FindAllMarkers" function in R package Seurat as described above.

To assess the gene expression signatures or pathway associated with the infiltration of $FAP^+$ fibroblasts and $SPP1^+$ macrophages, we first divide the TCGA-COAD cohort into four groups ($FAP^{High} SPP1^{High}$; $FAP^{High} SPP1^{low}$; $FAP^{Low} SPP1^{High}$; $FAP^{Low} SPP1^{Low}$) based on the above dichotomy. We then used GSEA[91] (http://software.broadinstitute.org/gsea/index.jsp) to test whether any hallmark gene sets are significantly enriched in the $FAP^+$ High $SPP1^+$ High group. We considered that gene signatures or pathways with FDR < 0.05 as significantly enriched.

**Quantitatively characterizing cell-cell communications**. NicheNet was used to infer the interaction between $FAP^+$ fibroblasts and $SPP1^+$ macrophages[61]. For ligands and receptor interactions, genes which are expressed in larger than 10% cells of clusters were considered. Top 100 ligands and top 100 targets of differential expressed genes of "sender cells" and "affected cells", were extracted for paired ligand-receptor activity analysis. When evaluating the regulatory network of $FAP^+$ fibroblasts on $SPP1^+$ macrophages, $THBS1^+$ macrophages were considered as reference receiver cells due to their potential differentiation trajectory to $SPP1^+$ macrophages. Meanwhile, $FGFR2^+$ fibroblasts and $ICAM1^+$ telocytes were used as reference cells to check the regulatory potential of $SPP1^+$ macrophages on $FAP^+$ fibroblasts. Nichenet_output$ligand_activity_target_heatmap was used to plot Ligands regulatory activity. Activity scores ranged from 0 to 1. The expression of differential expressed ligands and receptors were also shown in heatmap by calculating the average genes expression in indicated cell types and scaled across indicated subtypes.

**Spatial transcriptomics data analysis**. Spatial Transcriptomics (ST) slides were printed with two identical capture areas from four CRC patients. The capture of gene expression information for ST slides was performed by the Visium Spatial platform of 10x Genomics through the use of spatially barcoded mRNA-binding oligonucleotides in the default protocol. Raw sequencing reads of spatial transcriptomics were quality checked and mapped by Space Ranger v1.1. The gene-spot matrices generated after ST data processing from ST and Visium samples were analyzed with the Seurat package (versions 3.2.1) in R. Spots were filtered for minimum detected gene count of 200 genes while genes with fewer than 10 read counts or expressed in fewer than 3 spots were removed. Normalization across spots was performed with the LogVMR function. Dimensionality reduction and clustering were performed with independent component analysis (PCA) at resolution 1.1 with the first 30 PCs. Signature scoring derived from scRNA-seq or ST signatures was performed with the AddModuleScore function with default parameters in Seurat. Spatial feature expression plots were generated with the SpatialFeaturePlot function in Seurat (versions 3.2.1).

**Flow cytometry analysis of human cells**. Briefly, fresh human cells from normal mucosa and tumor tissue were washed and incubated with Live/Dead dye (Fixable Viability stain 520, BD Biosciences) in PBS for 10 min at 4 °C. After incubation, cells were washed in PBS with 2% FBS and 2 mM EDTA (FACS buffer). Fc block was performed for MSCs staining for 15 min at 4 °C, and not for myeloid cells. Indicated antibodies for MSCs or myeloid cells were diluted in FACS buffer with appropriate concentrations. Cells were stained for 30 min at 4 °C, and then washed and resuspended with FACS buffer. For stromal subsets analysis, the following antibodies were used: Alexa Fluor 488 anti-human CD326 (EpCAM) (clone 9C4), Biolegend, Cat#324210, dilution 1:200; Alexa Fluor 488 anti-human CD31 Antibody (clone WM59), Biolegend, Cat#303110, dilution 1:200; Alexa Fluor 700, anti-human CD45 Monoclonal Antibody (2D1), eBioscience, Cat#56-9459-42, dilution 1:200; PerCP/Cyanine5.5 anti-human CD325 (N-Cadherin) (clone 8C11), Biolegend, Cat#350814, dilution 1:200; Brilliant Violet 510 anti-human CD146 (clone P1H12), Biolegend, Cat#361022, dilution 1:200; Brilliant Violet 785 anti-human CD90 (Thy1) (clone 5E10), Biolegend, Cat#328142, dilution 1:200; Brilliant Violet 421 anti-human CD24 Antibody (clone ML5), Biolegend, Cat#311122, dilution 1:200; PE/Cyanine7 anti-human CD73 (Ecto-5'-nucleotidase) (clone AD2), Biolegend, Cat#344010, dilution 1:200; PE anti-human CD142 (clone HTF-1), eBioscience, Cat#12-1429-42, dilution 1:200; Alexa Fluor 700 anti-human ICAM1 (clone HA58), eBioscience, Cat#56-0549-42, dilution 1:200; BV605 anti-human CD26 (clone L272), BD OptiBuild, Cat#745244, dilution 1:200; APC anti-human FAP (clone # 427819), RD System, Cat# FAB3715A-100, dilution 1:200. For myeloid subsets analysis, the following antibodies were included, BV711 anti-human CD3 (clone UCHT1), BD Horizon, Cat#563725, dilution 1:200; BV786 anti-human CD19 (clone HIB19), BioLegend, Cat#302240, dilution 1:200; BV510 anti-human CD16b (clone CLB-gran11.5) (RUO), BD OptiBuild, Cat#744968, dilution 1:200; APC-Fire700 (APC-Cy7) anti-human XCR1 (clone S15046E), BioLegend, Cat#372608, dilution 1:200. APC anti-human CD1C (clone L161), BioLegend, Cat#331524, dilution 1:200. PE/Cyanine7 anti-human CD209 (clone 9E9A8), BioLegend, Cat# 330114, dilution 1:200; Alexa Fluor® 700 anti-human CD14 (clone HCD14), BioLegend, Cat#325614, dilution 1:200; Brilliant Violet 605™ anti-human CD45 (clone HI30), BioLegend, Cat#304042 PE anti-human CD13 (clone WM15), BioLegend, Cat#301704, dilution 1:200; PercP-Cy5.5 anti-human CD163 (clone GHI/61), BD Pharmingen, Cat#563887, dilution 1:200; PE-TexaRed (PE-CF594) anti-human CD206 (clone 19.2), BD, Cat#564063, dilution 1:200. Flow cytometry was performed on a BD Symphony (BD Biosciences) and obtained data by BD FACSDiva software v8.0.2, then analyzed with FlowJo v.10.5.3 (Tree Star Inc.). Statistical analysis was done by two-sided paired $t$-test with GraphPad Prism 6. $p < 0.05$ was considered as significant. ns, not significant. *$p < 0.05$, **$p < 0.01$, ***$p < 0.001$, ****$p < 0.0001$.

**Immunofluorescence staining**. Tissues were fixed in 1% PFA at 4 °C overnight, dehydrated with 30% sucrose over 12 h, and transferred to OCT and frozen in −80 °C for use. Tissues were sectioned into 10 μm-slices and rehydrated in PBS for 10 min. Permeabilization was done by soaking slices into pre-cooled methanol for 30 min at −20 °C. Sections were blocked with blocking buffer (0.3% Triton X-100, 1% BSA, 1% FBS and 0.1 mol/L Tris-HCL buffer) supplemented with goat and mouse serum. Rabbit anti-human FAP antibody (1:150) was stained for 3 h at room temperature and washed with washing buffer. Goat anti-rabbit AF647 (1:200), PE-labeled mouse anti-human osteopontin (SPP1), and Alexa Fluor 488-labeled mouse anti-human EPCAM was stained at 4 °C overnight in humid atmosphere. After washing, sections were mounted with anti-fading reagent and coated with coverslips. Images were observed with Leica microscopy and analyzed with Imaris Version 7.2.3.

**Chemerin concentration detection**. Concentration of chemerin was detected by enzyme-linked immunosorbent assay (ELISA) kit (R&D Systems, Cat.# DY2324) in accordance with the manufacturer's instructions. In summary, capture antibody was coated in 96-well plates overnight and washed three times with wash buffer (PBS with 0.05 % Tween 20) for three times. Plates were further blocked with 1% BSA in PBS (reagent dilution buffer) for 1 h and washed for three times. Chemerin standard concentration was done by two-fold dilution, and plasma was diluted at appropriate dilutions in regent dilution buffer from healthy donors and CRC patients. Incubation took 2 h at room temperatures and then washed. Chemerin was further detected by detection antibody, and incubated with HRP-conjugated antibody. TMB was used for reaction and 2N of $H_2SO_4$ was used to quench the reaction. Optical density (OD) was detected at 450 nm and 570 nm. Concentration of Chemerin was calculated with four-parameter logistic regression using the values of OD450 minus OD570. Unpaired $t$-test was used for statistical analysis. $p < 0.05$ was statistically significant. *$p < 0.05$.

**Reporting summary**. Further information on research design is available in the Nature Research Reporting Summary linked to this article.

## Data availability
The raw data of single-cell RNA-seq and spatial transcriptomics generated in this study were deposited in Genome Sequence Archive with accession ID HRA000979. Since these

data are related to human genetic resources, raw data can be obtained within half year by requesting and following the guidelines for Genome Sequence Archive for non-commercial use at https://ngdc.cncb.ac.cn/gsa-human/request/HRA000979. There are no time restrictions once access has been granted. The guidance for making a data access request of GSA for humans can be downloaded from https://ngdc.cncb.ac.cn/gsa-human/document/GSA-Human_Request_Guide_for_Users_us.pdf. The processed gene expression data is submitted as Supplementary Data 3 and the metadata is submitted as Supplementary Data 4. The processed CRC public scRNA-seq dataset were downloaded from Gene Expression Omnibus (GEO, https://www.ncbi.nlm.nih.gov/geo/), including GSE146771[16], GSE132465[14], and GSE144735[14]. The public normalized gene expression data based on fragments per kilobase of exon model per million reads mapped (FPKM) of colon adenocarcinoma and rectum adenocarcinoma were obtained from TCGA data portal (http://gdac.broadinstitute.org/). Expression datasets based on Affymetrix microarray of CRC patients were downloaded from GEO, including GSE41568[39], GSE39582[38], GSE37892[37], GSE33113[36], GSE21510[34], GSE20916[33], GSE18105[32], GSE13294[29], GSE14333[30], GSE23878[35], GSE17536[31], and GSE17537[31]. The data information (e.g., sample size, overall survival times, progressive free survival time) were summarized in Supplementary Table 2. The value of infiltration of lymphocyte, the richness of T-cell receptor, non-silent mutation rate, and neoantigen load of CRC were obtained from Thorsson et al.[91]. The public gene expression data and detailed clinical information for patients with anti-PD-L1 (antibody: atezolizumab) treatment were obtained from IMvigor210 cohort (Intervention treatment of advanced urinary tract transitional cell carcinoma[67]). Source data are provided with this paper.

## Code availability

Codes were implemented in R 3.6.0 and are deposited in https://github.com/youqiongye/CRC_scRNAseq/tree/V1.0.0.

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

## Acknowledgements

This work was supported by grants from the National Key Research and Development Program of China (2021YFA1301400), and National Natural Science Foundation of China (No. 82073145, 31930035, 91942311, and 81971487). Shanghai Science and Technology Commission (20JC1410100), Shanghai Pujiang Program (No. 20PJ1412800), and Natural Science Foundation of Shanghai (No. 20ZR1472900).

## Author contributions

Y.Y and B.S. conceived and supervised the project. J.Q., Y.Y., and B.S. designed and performed the research. Y.Z., Z.W., J.Z., D.Z., L.L., and Y.L. collected the CRC samples. H.S., Z.X., X.D., R.B., L.C., J.Q., and Y.Y. performed the data analyses. J.Q., L.Hong, W.J., F.F., and H.L. performed experiments. J.Q., H.S., Z.L., L.Han, F.G., Y.Y., and B.S. interpreted the results. J.Q., H.S, Y.Y., and B.S. wrote paper with input from all the other authors.

## Competing interests

The authors declare no competing interests. A patent titled "Biomarker panel and application of cell subpopulation of tumor boundary in Colorectal cancer" related to this work is under application and currently has no identification number. The authors declare non-financial competing interests in this patent.
