## [Peer Review File · Nature Communications]

Single-cell and spatial analysis reveal interaction of FAP+ fibroblasts and SPP1+ macrophages in colorectal cancerReviewers' Comments:

Reviewer #1:

Remarks to the Author:

In the manuscript entitled "Single-cell transcriptomic profiling of human colorectal cancer reveals a cross-talk of tumor-specific FAP+ fibroblasts and MARCO+ macrophages", Qi and colleagues provide single cell RNA sequencing data and careful analysis of 5 colorectal cancers and matched normal tissues. They focus on cells belonging to the tumor microenvironment (TME) and very nicely integrate their data with existing single cell and bulk RNAseq datasets to uncover associations in abundance among different stromal/immune cells. They also conduct a number of creative computational analyses to generate hypotheses about function and interaction of cells in the TME.

This is a high-quality manuscript that could essentially be published as is. However, I will provide a few suggestions that the authors can consider to further increase the impact of their work.

Major comments:

The authors should probably be a bit more careful with some of their claims. Repeatedly, they suggest that particular cells are causally implicated in tumor progression or the tumor's clinical behavior based on association with outcome, but association does not necessarily suggest causality. More careful wording would be appreciated throughout the manuscript. Also, the title makes a strong claim ("cross-talk between cells") which is not experimentally proven. I don't think associations or computational interaction analyses are sufficient to back such statements. No additional work is needed here – language adjustments will suffice.

The manuscript is very long. I appreciate the level of detail, but the length may reduce its impact a bit. Perhaps the authors could consider moving the detailed descriptions of how their cell types relate to other published cell types (in other single cell data sets) to a supplementary note.

There is no information on how well CIBERSort does in decomposing bulk data into cell types. Can the authors report some performance metrics and show that expression signatures of their cell types are sufficiently distinct that the correlations observed in Fig. 2C do not simply reflect correlation of gene expression (but instead really correlation of cell type abundance)?

The poor prognosis of mesenchymal-type CRCs is well known (Guinney Nature Med 2015). Can the authors comment on to what degree their analysis in Fig. 2D/E is simply "rediscovering" that effect? This is not a criticism that takes away from the novelty here, it might just be nice to add this for purposes of academic rigor. The authors integrate their data extensively with published single cell RNAseq data sets – can they attempt the same for the known bulk expression subtypes? I think this would be really helpful for the field.

The colocalization analysis in Fig. 5D is very nice, but it requires a quantification across tumors (or at least multiple fields of view) to prove generality.

Minor comments:

Line 172-175: I do not see how enrichment of these pathways suggests strong cell-cell interaction. Could you please elaborate or rewrite? Similarly, I do not see how upregulation of hypoxia gene sets in a number of cell types indicates a cell-interaction network. This is related to the first point of the major comments.

Can the authors clarify in the figure legend of Fig. 1B whether the two UMAP plots are derived from the same analysis (and the cells from tumor/normal tissue were simply plotted separately afterwards) or whether they represent two different analyses ?

Can the authors add epithelial cells to Figure 2A? They may not be primarily interested in them, but this would be helpful for readers with other interests. Or are they not comparable because of the EDTA treatment? Can you explain why you chose to treat tumors vs. normal differently?

Line 407 – “to investigate xxx”???

Line 415 – enhanced with respect to what?

Reviewer #2:

Remarks to the Author:

Qi and colleagues propose a potential interaction between FAP+ fibroblasts and MARCO+ macrophages as an important culprit of malignant features in colorectal cancer. No doubt, this work contains interesting clues regarding the role of these subsets but it has been extremely difficult for this Reviewer to evaluate the authors' work. While well-written from a language point of view, it was very difficult to understand the logic of some of the author's conclusions which has been shadowed throughout the paper with consecutive interrogations of the data or purely descriptive narratives with little biological content. I think the paper has to be rewritten in order to be properly evaluated.

Other major comments:

1. The premise of the paper is “Insufficient knowledge of changes in the intestinal tissue during the transformation to cancer limited quick development of novel preventive therapeutic targets on non-T immune cells and other cell types, such as stromal cells, in the tumor microenvironment.” – this is true, however, the authors have not addressed this problem as they compare normal mucosa with full blown cancer. To evaluate processes of transformation one would need to look at earlier stages of malignant transformation. Moreover, CRC is very heterogeneous and unlikely to be well-represented by only 5 samples.
2. I noticed the authors have included 2 rectal cancers in their analysis. According to the guidelines, rectal cancers receive neoadjuvant therapy prior to resection and, therefore, I suspect that those two cases would have specific features. This has not been addressed by the authors.
3. Figure 1 - How was it possible to generate UMAPs that look exactly the same between tumor and normal samples? Or was just one UMAP constructed and then cells from both type of samples are selectively presented? This Reviewer finds it striking that there are no populations that specifically cluster in normal or tumor samples. I find it very unlikely that epithelial cells derived from normal and tumor tissues will occupy the same space in UMAP embedding. As such, I do not think the UMAP figure presented in Figure 1 actually represents the data generated by the authors.
4. Regarding figure 2A, the hallmark gene sets were mainly developed to investigate transcriptional alterations occurring in tumor tissues (bulk), not really to be applied to immune cell subsets at single-cell level. Why investigate UV response in (immune) single cells derived from CRC? Or epithelial to mesenchymal transition in lymphocytes? The authors should focus on comparing transcriptional signatures of different subsets between normal and tumor tissues. It is fine if they wish to do this at pathway level but then they should employ more logical gene sets where the biology makes sense.
5. “Microenvironment remodeling-related pathways, including angiogenesis, apical junction, hypoxia, and epithelial-mesenchymal transition, were enriched in both tumor stromal cells and immune cells (Figure 2A), suggesting strong cell-cell interaction in the TME that could promote tumorigenesis.” I am not sure whether the authors are referring to physical interaction or paracrine interaction. Many of these pathways are regulated by growth factors that will produce similar effects in terms of

signaling and transcriptional activity in various subsets. This, however, does not imply cell-cell interaction in a physical sense but more of an overall effect of signaling molecules. In general too many assumptions are made based on this analysis while the great majority of pathways is significantly enriched in tumor-derived subsets. This type of side-steps in the analysis make it very difficult to accompany the overall message of the paper.

6. Section – “Alteration of cell infiltration between normal and tumor tissue support the remodeling of tumor microenvironment” in CRC please write this section, from the start, from a comparative perspective instead of listing the individual characteristics of each subset.

7. I am puzzled and surprised by the following observation: “dramatic decrease of memory CD4+ T cells.” If I am not mistaken this goes against expectations and what has been generally described in CRC. Memory CD4+ T cells should be more abundant in tumors. The authors should be careful regarding these comparisons as they are simply looking at the proportion of subsets.

8. In general there is an abusive use of CIBERSORTx to investigate particular subsets in bulk RNA datasets without prior validation of the capacity of CIBERSORTx to indentify those subsets. This makes the conclusions based on large cohorts rather weak.

9. In general, the reported associations between stromal content or myeloid infiltration with poorer clinical features in CRC are already known

Minor:

Some phrasing, particularly in the introduction deserves some attention:

“Colorectal cancer (CRC) is one of the most common malignancies involving a complex interplay among different cell types via multiple physical and chemical factors with no available non-surgical therapies effective for the whole wide spectrum of colorectal cancer sub-types.”

This problem is defined as being specific for colorectal cancer but, I think we can all agree that there will not be a non-surgical effective therapy for the whole wide spectrum of any cancer type. Please replace the last part of the sentence for a more realistic statement.

“Most previous studies focused on the generation of effective anti-tumor responses by T cells [5].” Unclear what this refers to but I am confident that all-in-all more non-immunotherapeutic avenues have been explored, to date, than immunotherapeutic ones.

“In addition, infiltrating immune cells excluded TME has been shown to be associated with CRC.” Rephrase.

Reviewer #3:

Remarks to the Author:

This manuscript studies the tumor microenvironment of colorectal cancer (CRC), particularly one two very interesting subsets of fibroblasts and macrophages. The authors focus primarily on single cell RNA sequencing (scRNAseq) of 5 patients, profiling both the tumor as well as juxta-tumor tissue. The scRNAseq data is of high quality and the authors perform state of the art, comprehensive analyses. Hypotheses generated from this scRNAseq cohort are validated in turn with multiple public cohorts of scRNAseq and bulk RNAseq, immunohistochemistry and FACS. All of this is used to build a compelling model of the mechanisms generating these two tumor-specific non-malignant populations, the interactions between them, and their association with T cells, tumor development, and immune checkpoint inhibitor response.

The study is very well done and is appropriate for the Nature Communications audience. I have no major concerns but I detail below a number of minor suggestions that might improve the paper.

Minor Comments:

1. English should be reviewed carefully for word choice, typos, and phrasing.
2. Term "normal" tissue should be replaced by "juxta" or "adjacent" tissue as much as possible.
3. Why are there no UMAPs for endothelial or glial markers in fig 1D?
4. Throughout the manuscript there are many Kaplan Meier curves and analyses of overall survival (OS) and progression free survival (PFS). In all cases the authors use the `maxstat.test` function to split the cohort into high and low for the quantitative explanatory variable (eg CIBERSORTx estimate of cellular proportion). This is statistically acceptable but can be susceptible to small variations in the distributions. It would be nice to add in the supplemental Kaplan Meier analyses split just by median and/or the actual distributions of the explanatory variables so the reader can estimate how robust these survival analyses are with respect to the cutpoint.
5. Line 407 - "To investigate xxx" must be a typo
6. Fig 4D is difficult to gauge visually. Maybe an alternative layout, like a violin plot or the actual distributions would make it easier to estimate if the bulk of the distribution is shifted or just a few outliers.
7. Need at least some information about the cohorts for Fig 6D to gauge how well matched they are, minimally age and gender.
8. Sentence from line 502-504 seems to be incomplete.
9. Fig 7A-E show a very large range for many of the boxes - the actual density and variation might be better visualized by showing the individual points instead of box and whisker.
10. Why aren't there 4 curves for fig 7H,I (FAPhigh/low, MARCOhigh/low)?

Point-by point Responses to Reviewers' Comments:

Reviewer #1, expert in colorectal cancer genomics (Remarks to the Author):

In the manuscript entitled “Single-cell transcriptomic profiling of human colorectal cancer reveals a cross-talk of tumor-specific FAP⁺ fibroblasts and MARCO⁺ macrophages”, Qi and colleagues provide single cell RNA sequencing data and careful analysis of 5 colorectal cancers and matched normal tissues. They focus on cells belonging to the tumor microenvironment (TME) and very nicely integrate their data with existing single cell and bulk RNAseq datasets to uncover associations in abundance among different stromal/immune cells. They also conduct a number of creative computational analyses to generate hypotheses about function and interaction of cells in the TME.

This is a high-quality manuscript that could essentially be published as is. However, I will provide a few suggestions that the authors can consider to further increase the impact of their work.

Response: We thank the reviewer for these positive comments that highlight the significance of our study.

Major comments:

The authors should probably be a bit more careful with some of their claims. Repeatedly, they suggest that particular cells are causally implicated in tumor progression or the tumor’s clinical behavior based on association with outcome, but association does not necessarily suggest causality. More careful wording would be appreciated throughout the manuscript. Also, the title makes a strong claim (“cross-talk between cells”) which is not experimentally proven. I don’t think associations or computational interaction analyses are sufficient to back such statements. No additional work is needed here – language adjustments will suffice.

Response: We agree and thank this reviewer for these valuable comments. We have revised our conclusions including the title and changed the word “cross-talk” to “direct interaction” in the revised manuscript to avoid overstatement. In revision, we also performed additional immunofluorescence and spatial transcriptomics (ST) analysis to support the interaction of FAP⁺ fibroblasts and MARCO⁺ macrophages.

The manuscript is very long. I appreciate the level of detail, but the length may reduce its impact a bit. Perhaps the authors could consider moving the detailed descriptions of how their cell types relate to other published cell types (in other single cell data sets) to a supplementary note.

Response: Thanks for the reviewer’s suggestion. We have moved “**Comparison of myeloid cell subtypes with previous publication**” and “**Comparison of MSCs subtypes with previous publication**” these two panels in the original manuscript to “**Supplementary Results**”, respectively, in revision.

In addition, we have moved the section “**Alteration of cell infiltration between adjacent normal and tumor tissue support the remodeling of tumor microenvironment in CRC**” in the original manuscript to “**Supplementary Results**” in revision.

There is no information on how well CIBERSort does in decomposing bulk data into cell types. Can the authors report some performance metrics and show that expression signatures of their cell types are sufficiently distinct that the correlations observed in Fig. 2C do not simply reflect correlation of gene expression (but instead really correlation of cell type abundance)?

Response: We used CIBERSORTx but not CIBERSORT in our study. CIBERSORT uses known cell-specific bulk transcriptomics as reference gene expression signatures to characterize cell composition of bulk tissues (Newman et al., 2015), while CIBERSORTx allows the use of single-cell RNA-sequencing data as reference gene expression signatures to analyze cell type composition of bulk tissues from their gene expression profiles (Newman et al., 2019).

As suggested, we compared concordance between cell type composition determined by CIBERSORTx deconvolution and the true cell type composition determined by scRNA-seq from 3 held-out CRC tumors to evaluate the robustness of the performance of CIBERSORTx (Newman et al., 2019). In brief, we created a signature matrix from a training cohort consisting of two CRC specimens in our scRNA-seq dataset and used the remaining three CRC specimens as the test dataset (**Supplementary Fig. 1a**, in revision). As shown in **Fig. 1b** in revision, the signature matrix distinguished nicely the epithelial cells, MSCs, endothelial cells, glial cells, B cells, plasma cells, T/ILCs, myeloid cells, and mast cells. When evaluated using reconstructed three CRC samples, deconvolution results were highly concordant with ground truth cell proportions determined by scRNA-seq (**Supplementary Fig. 1b**, in revision). Furthermore, scRNA-seq and CIBERSORTx proportions for 9 cell types in 3 held-out CRC specimens were also well correlated ($R_s = 0.93$, $p = 1.2 \times 10^{-12}$; **Supplementary Fig. 1c**, in revision). In addition, the deconvolution performance of CIBERSORTx has been validated previously on simulated tumors reconstructed from single cell RNA-seq data, including melanoma, head and neck squamous cell carcinoma, and non-small cell lung cancer (Newman et al., 2019).

We added these results as the new section “**Validation of CIBERSORTx performance**” in **Supplementary Results** in revised manuscript.

The poor prognosis of mesenchymal-type CRCs is well known (Guinney Nature Med 2015). Can the authors comment on to what degree their analysis in Fig. 2D/E is simply “rediscovering” that effect? This is not a criticism that takes away from the novelty here, it might just be nice to add this for purposes of academic rigor. The authors integrate their data extensively with published single cell RNAseq data sets – can they attempt the same for the known bulk expression subtypes? I think this would be really helpful for the field.

Response: We thank this reviewer for this kind suggestion. Based on the bulk RNA-seq and RNA microarray, Guinney *et al.* constructed the consensus molecular subtypes (CMS) to categorize colorectal cancer, and identified one subtype CMS4 (Mesenchymal, 23%) that was featured by prominent transforming growth factor β activation, stromal invasion, and angiogenesis (Guinney et al., 2015). Importantly, this mesenchymal-type CRC showed worse relapse-free and overall survival. In our **Fig. 2d-e**, we not only showed the correlation of high infiltration of MSCs in CRC with worse overall survival (OS) and progression-free survival (PFS)

rate of CRC patients, we also provided a critical link between the stromal and macrophage in this class of CRC, highlighting the potential significance of interaction of stromal subtype *FAP*⁺ fibroblast and myeloid cell subtype *MACRO*⁺ macrophage. We believe that this is at least one of the mechanisms to explain the poor outcome for the patients. This hypothesis is supported not only by our imaging data and single cell RNA-seq data but also from the newly generated spatial transcriptomics data. We have discussed this with the new data in revised manuscript (page 15-16).

Furthermore, we compared the cell type composition determined by CIBERSORTx deconvolution in the known bulk expression subtypes in TCGA CRC cohort. We found that both MSCs and myeloid cells showed higher infiltration in CMS4 (**Supplementary Fig. 2i-j**, in revision) than other CMS types. We also uncovered that the infiltration of MSCs was positively related to high infiltration of myeloid cells (**Fig. 2b-c**), suggesting that this type of CRC (CMS 4) might accompanied by high infiltration of myeloid cells. Discussion of our study and that by Guinney *et al.* are added in the revised manuscript (page 8).

The colocalization analysis in Fig. 5D is very nice, but it requires a quantification across tumors (or at least multiple fields of view) to prove generality.

Response: As suggested, to better quantitate cell-cell interaction of *FAP*⁺ fibroblasts and *MARCO*⁺ macrophages, we generated new data from immunofluorescence study and spatial transcriptomics study. We found that *SPP1* was specifically expressed in *MARCO*⁺ macrophages from scRNA-seq data (Supplementary Table 4). Luckily, there is a commercially available anti-*SPP1* antibody working well in immunofluorescence assay, we further confirmed and also quantified the colocalization of *FAP*⁺ fibroblasts and *MARCO*⁺ macrophages using anti-*SPP1* antibody (**Fig. 5d-e**, in revision). Description of new figure **Fig. 5d-e** was added to revised manuscript (page 15).

Minor comments:

Line 172-175: I do not see how enrichment of these pathways suggests strong cell-cell interaction. Could you please elaborate or rewrite? Similarly, I do not see how upregulation of hypoxia gene sets in a number of cell types indicates a cell-interaction network. This is related to the first point of the major comments.

Response: We agree and thank the reviewer for pointing out this overstatement. We re-phrased it as “Tissue remodeling-related pathways, including angiogenesis, apical junction, UV response, hypoxia, and epithelial-mesenchymal transition, were enriched in both tumor stromal cells and immune cells (**Fig. 2a**), suggesting these two cell types act synergistically to promote tumorigenesis” in revised manuscript (page 7).

Can the authors clarify in the Fig. legend of Fig. 1B whether the two UMAP plots are derived from the same analysis (and the cells from tumor/normal tissue were simply plotted separately afterwards) or whether they represent two different analyses?

Response: We are sorry for this confusion. The two UMAP plots are derived from the same analysis using R package *harmony* to correct batch effects and constructed one UMAP based on all cells from adjacent normal tissue and tumor, then split cells by these two tissues. We clarified this in figure legend of **Fig. 1b** in revised manuscript.

Can the authors add epithelial cells to Fig. 2A? They may not be primarily interested in them, but this would be helpful for readers with other interests. Or are they not comparable because of the EDTA treatment? Can you explain why you chose to treat tumors vs. normal differently?

Response: We mainly focused on the non-epithelial cell compartment of CRC (MSCs and immune cells). The EDTA treatment in adjacent tissue could greatly reduce the epithelial cells number in total cell suspensions, but still we could get information of the differential expressed genes between tumor and adjacent tissue. The adjacent tissues contain quite a lot of epithelial cells, and will decrease the stromal cells and immune cells compartment of our interest. Therefore, we used the EDTA to dissociate most of the epithelial cells in adjacent tissue (Martin et al., 2019). Additionally, removing the epithelial layer would benefit the digestion of the remaining tissue in terms of cell viability and diversity. However, for the tumor tissues, since the epithelial cells were usually within the tumor and would not be removed by EDTA treatment. Therefore, we treated differently in an effort to get maximum number of non-epithelial cells.

Line 407 – “to investigate xxx”???

Response: We are sorry for this mistake, and we have corrected it as “We investigated the alterations in myeloid cell subtypes among adjacent tissues and tumor tissues” in the revised manuscript.

Line 415 – enhanced with respect to what?

Response: We are sorry for this confusion, and we have corrected it as “Consistent with these results, the infiltration of *MARCO*⁺ macrophages in tumor samples was found to be significantly increased as compared with that in adjacent normal tissues when evaluated the cell infiltration of the TCGA cohort by CIBERSORTx” in revised manuscript (page 12).

Reviewer #2, expert in immunogenomics and immunotherapies for colorectal cancer (Remarks to the Author):

Qi and colleagues propose a potential interaction between FAP⁺ fibroblasts and MARCO⁺ macrophages as an important culprit of malignant features in colorectal cancer. No doubt, this work contains interesting clues regarding the role of these subsets but it has been extremely difficult for this Reviewer to evaluate the authors’ work. While well-written from a language point of view, it was very difficult to understand the logic of some of the author’s conclusions which has been shadowed throughout the paper with consecutive interrogations of the data or purely descriptive narratives with little biological content. I think the paper has to be rewritten in order to be properly evaluated.

Response: We thank the reviewer for raising this concern and we have tried our best to re-write in order to clearly state our results and conclusions in the revised manuscript. We re-organized the manuscript as follows to make the logic clear.

1. We kept our main story line: 1) we dissected the tumor microenvironment between CRC tissues and adjacent normal tissues at a single-cell resolution; 2) we found that tumor-specific *FAP*⁺ fibroblasts and tumor-specific *MARCO*⁺ macrophages were associated with colorectal cancer progression; 3) we assessed the association of *FAP*⁺ fibroblasts and *MARCO*⁺ macrophages through integrated analysis of scRNA-seq data from this study, and publicly accessible scRNA-seq and bulk RNA-seq datasets; 4) we validated the physic interaction of *FAP*⁺ fibroblasts and *MARCO*⁺ macrophages by immunofluorescent imaging studying and **spatial transcriptomics (see response to Reviewer #2's major comment #5)**; 5) *FAP*⁺ fibroblasts and *MARCO*⁺ macrophages interaction may contribute to desmoplastic tumor microenvironment; 6) high infiltration of *FAP*⁺ fibroblasts and *MARCO*⁺ macrophages correlate with worse patient survival and immunotherapy resistance.
2. We moved three panels with branch and supporting contents to **Supplementary Results**: 1) the interpretation of subtypes of epithelial cells, endothelial cells, T cells, and B cells, between adjacent normal tissues and tumor tissues (the section “Alteration of cell infiltration between normal and tumor tissue support the remodeling of tumor microenvironment in CRC” in the original submission); 2) validation of CIBERSORTx performance (a new section); 3) comparison of cell types and other published studies (the section “Comparison of MSCs subtypes with data from previously published data” and “Comparison of myeloid cell subtypes with previous publication” in the original submission).
3. We have sent our manuscript to WOSCIENCE (<http://www.wosci.cn>) for English-language editing to improve the grammar and sentence structure. Changes are marked with red label.

Other major comments:

1. The premise of the paper is “Insufficient knowledge of changes in the intestinal tissue during the transformation to cancer limited quick development of novel preventive therapeutic targets on non-T immune cells and other cell types, such as stromal cells, in the tumor microenvironment.” – this is true, however, the authors have not addressed this problem as they compare normal mucosa with full blown cancer. To evaluate processes of transformation one would need to look at earlier stages of malignant transformation. Moreover, CRC is very heterogeneous and unlikely to be well-represented by only 5 samples.

Response: We are sorry for this confusion. In our study, we didn't focus on the processes of cancer transformation, therefore we revised this sentence as “Insufficient knowledge about the changes in the intestine between adjacent normal and tumor tissues has prevented urgent identification of novel preventive therapeutic targets of non-T immune cells, such as stromal cells, in the tumor microenvironment (TME)” in the introduction in our revised manuscript (page 3).

We agree completely that “CRC is very heterogeneous and unlikely to be well-represented by only 5 samples”. We indeed integrated our dataset from 5 samples with two previous studies

(**Supplementary Fig. 3 and 5**) to validate our results. As the sample size of the scRNA-seq dataset was limited, we applied the deconvolution software CIBERSORTx to cell type-specific gene expression profiles from our scRNA-seq to estimate the composition of cell types in large datasets from the CRC cohort of The Cancer Genome Atlas (TCGA) and 12 independent CRC cohorts from Gene Expression Omnibus (GEO). The imputation analysis highlighted the interaction of *FAP*⁺ fibroblasts and *MARCO*⁺ macrophages, and its correction with resistance of immunotherapy. Furthermore, we performed integrated analysis of spatial transcriptomics, FACS, and IF to validate our results. Our findings may lead to the development of novel therapeutic intervention by disrupting the interaction of *FAP*⁺ fibroblasts and *MARCO*⁺ macrophages to increase the infiltration of immune cell in tumor core. Furthermore, we added additional spatial transcriptomics data to support our conclusion in revised manuscript (page 15-16). We also discussed the limitations of our analysis on the clinical trials in the revised manuscript (page 23).

2. I noticed the authors have included 2 rectal cancers in their analysis. According to the guidelines, rectal cancers receive neoadjuvant therapy prior to resection and, therefore, I suspect that those two cases would have specific features. This has not been addressed by the authors.

Response: We thank the reviewer for asking this question. According to the guidelines in China, patients first diagnosed with resectable rectal cancer do not need to receive neoadjuvant therapy prior to resection (National Health Commission of the People's Republic of, 2020). The two rectal patients included in our study did not receive neoadjuvant therapy prior resection.

3. Fig. 1 - How was it possible to generate UMAPs that look exactly the same between tumor and normal samples? Or was just one UMAP constructed and then cells from both type of samples are selectively presented? This Reviewer finds it striking that there are no populations that specifically cluster in normal or tumor samples. I find it very unlikely that epithelial cells derived from normal and tumor tissues will occupy the same space in UMAP embedding. As such, I do not think the UMAP Fig. presented in Fig. 1 actually represents the data generated by the authors.

Response: We are sorry for this confusion. The two UMAP plots of **Fig. 1b** are derived from the same analysis using R package *harmony* to correct batch effects and constructed one UMAP profile based on all cells from the adjacent normal tissue and tumor tissue. The data from each single cell of the two tissues were selectively projected onto the UMAP individually. We clarified this in the figure legend of **Fig. 1b** in revised manuscript (page 40).

UMAP in **Fig. 1b** represented an overall view of all cells from adjacent normal tissue and tumor tissue that were clustered into nine major subtypes based on unsupervised clustering. The individual cluster (*e.g.*, epithelial cell cluster, myeloid cell cluster) from this UMAP could be further clustered into different sub-clusters independently for more in-depth analyses. For example, epithelial cell cluster analysis revealed major difference between adjacent tissue and tumor tissues since adjacent normal tissues consisted mainly TA and stem-like cells, while tumor tissue mainly consisted of malignant cells, enterocyte 2, and proliferating TA (**Supplementary Fig. 2e-h**). Because this study mainly focused on MSCs and myeloid cells, only the composition (primary clustering) of epithelial cell, endothelial cells, B-, and T-cells between adjacent and

tumor tissues were presented (**Supplementary Fig. 2e-h**). We have discussed this panel in **Supplementary Results**, and added this as **Supplementary Fig. 2e-h** in revised manuscript.

4. Regarding Fig. 2A, the hallmark gene sets were mainly developed to investigate transcriptional alterations occurring in tumor tissues (bulk), not really to be applied to immune cell subsets at single-cell level. Why investigate UV response in (immune) single cells derived from CRC? Or epithelial to mesenchymal transition in lymphocytes? The authors should focus on comparing transcriptional signatures of different subsets between normal and tumor tissues. It is fine if they wish to do this at pathway level but then they should employ more logical gene sets where the biology makes sense.

Response: We thank the reviewer for this valuable comment. Indeed, we performed an unbiased gene set enrichment analysis across individual cell types between tumors and adjacent normal tissues using the 50 Molecular Signatures Database (MSigDB) hallmark gene sets (**Fig. 2a**), which comprised of 4,022 of the original 8,380 MSigDB gene sets. These gene sets include immune, metabolism, signaling, proliferation, DNA damage, and development signatures, representing well-defined biological processes. The UV response related gene set may also play important roles in immune cells (Maglio et al., 2016; Witkowski et al., 2020). Previous studies also reported that immune cells play roles in epithelial to mesenchymal transition (Reiman et al., 2010; Ricciardi et al., 2015). We discussed this in revised manuscript (page 7).

5. “Microenvironment remodeling-related pathways, including angiogenesis, apical junction, hypoxia, and epithelial-mesenchymal transition, were enriched in both tumor stromal cells and immune cells (Fig. 2A), suggesting strong cell-cell interaction in the TME that could promote tumorigenesis.”

I am not sure whether the authors are referring to physical interaction or paracrine interaction. Many of these pathways are regulated by growth factors that will produce similar effects in terms of signaling and transcriptional activity in various subsets. This, however, does not imply cell-cell interaction in a physical sense but more of an overall effect of signaling molecules. In general too many assumptions are made based on this analysis while the great majority of pathways is significantly enriched in tumor-derived subsets. This type of side-steps in the analysis make it very difficult to accompany the overall message of the paper.

Response: We agree that this statement is too strong as our current data could not directly prove that there is a cell-cell interaction in the TME. We thus revised this statement to “suggesting that these two cell types may act synergistically to promote tumorigenesis” in revised manuscript.

However, we did validate the physical interaction of *FAP*⁺ fibroblasts and *MARCO*⁺ macrophages by new IF assay (**the Reviewer #1’s major comment #5**) and by spatial transcriptomics study (**Fig. 6a and 6d**). Based on the unbiased clustering and spot features, all spots of CRC patient #6 were clustered into 10 spot clusters, including *FAP*⁺ fibroblasts/*MARCO*⁺ macrophages, malignant epithelial cells, epithelial cells, MSCs, *MT*⁺ with fibrosis, *FAP*⁺ fibroblasts, myofibroblast cells, endothelial cells, immune cells, and Unknown as shown in revised **Fig. 6b-c**. The spots of CRC patient #7 were clustered into 6 spot clusters, including *FAP*⁺ fibroblasts/*MARCO*⁺ macrophages, malignant epithelial cells, MSCs, myofibroblast cells, immune cells, and Unknown (**Fig. 6e-f**). Score the gene signatures of the

spots from scRNA-seq data (top 25 specifically expressed genes) in each cluster with *FAP*⁺ fibroblasts or *MARCO*⁺ macrophages highlighted a cluster with colocalization of *FAP*⁺ fibroblasts and *MARCO*⁺ macrophages, which wrapped around the malignant epithelial cells (**Fig. 6g-i**). In addition, the gene signature scores of *FAP*⁺ fibroblasts and *MARCO*⁺ macrophage were positively correlated (**Fig. 6j-k**). Most of *FAP*⁺ fibroblasts and *MARCO*⁺ macrophages colocalized to the same spots, and a spot in the 10x Genomics Visium platform could accommodate up to 10 cells, suggesting a possible physical interaction between the two cell types. Furthermore, we found that spots with colocalization of *FAP*⁺ fibroblasts and *MARCO*⁺ macrophages were highly enriched in pathways such as extracellular matrix organization, collagen fibril organization, and response to TGF- β , suggesting that they may contribute to the formation of desmoplastic structures (**Fig. 6l**). The infiltrating T cells or B cells appeared to be excluded from the tumor epithelium (**Fig. 6m**), suggesting that the desmoplastic microenvironment may be regulated by the interaction of *FAP*⁺ fibroblasts and *MARCO*⁺ macrophages to limit the infiltration of immune cells into the tumor core.

The quality control and supportive information are shown in **Supplementary Fig. 8**. We added a new section “Cell-cell interaction of *FAP*⁺ fibroblasts and *MARCO*⁺ macrophages revealed by spatial transcriptomics” in revised manuscript (page 14-15).

6. Section – “Alteration of cell infiltration between normal and tumor tissue support the remodeling of tumor microenvironment” in CRC please write this section, from the start, from a comparative perspective instead of listing the individual characteristics of each subset.

Response: We have re-organized this section and moved it to “**Supplementary Results**” in revised manuscript.

7. I am puzzled and surprised by the following observation: “dramatic decrease of memory CD4⁺ T cells.” If I am not mistaken this goes against expectations and what has been generally described in CRC. Memory CD4⁺ T cells should be more abundant in tumors. The authors should be careful regarding these comparisons as they are simply looking at the proportion of subsets.

Response: We thank the reviewer for this suggestion. The memory CD4⁺ T cells were featured by absence of IFN-g or IL-17A expression, and presence of CD4 expression, and positive for *SELL* and *CCR7* (Liu et al., 2020), and effector T were defined as *IFNG* positive and *IL17A* positive in the TEM population. The % of memory CD4⁺ T cells was calculated by its fraction in total T/ILCs cells, and its decreased percentage (still account for about 15% of total T cells) might be due to the relative increase of Treg and NK (**Fig. A1**). We also validate the change of memory CD4⁺ T cells between the adjacent normal tissue and tumor tissue using published single cell-transcriptomic data (Zhang et al., 2018), which showed that the percentage of memory CD4⁺ T in tumor was decreased as compared to that in adjacent normal tissue (**Fig. A1B**). We revised the sentence as “significant decrease of memory CD4⁺ T cells” as suggested in **Supplementary Results** in revised manuscript.

Fig. A1. Alteration of memory $CD4^+$ T cells between adjacent normal tissue and tumor tissue. (A) Dotted line graphs show percentages of each subtype among T/ILCs cells in adjacent mucosa and tumor tissues. Donors are color-coded. (B) Bar plot shows the percentage of memory $CD4^+$ T cells (red) and remained T cells (blue) between adjacent normal tissue and tumor. Statistical analysis is performed by paired two-side t-test in A and Fisher test in B.

8. In general there is an abusive use of CIBERSORTx to investigate particular subsets in bulk RNA datasets without prior validation of the capacity of CIBERSORTx to indentify those subsets. This makes the conclusions based on large cohorts rather weak.

Response: As mentioned in our response to the Reviewer #1's major comment #3, we have validated the robustness of the use of CIBERSORTx using our own scRNA-seq dataset from the five CRC patients based on the validation method provided by the developers of CIBERSORTx (Newman et al., 2019).

9. In general, the reported associations between stromal content or myeloid infiltration with poorer clinical features in CRC are already known

Response: Most previous studies reported the associations between stromal or myeloid cell infiltration with poorer clinical features in CRC by bulk RNA sequencing. These would not allow further analysis of the association of these cells. Here we used single cell RNA-seq data to investigate the association and potential interaction of stromal subtype FAP^+ fibroblast and myeloid cell subtype $MARCO^+$ macrophage for their potential roles in CRC.

Minor:

Some phrasing, particularly in the introduction deserves some attention:

“Colorectal cancer (CRC) is one of the most common malignancies involving a complex interplay among different cell types via multiple physical and chemical factors with no available non-surgical therapies effective for the whole wide spectrum of colorectal cancer sub-types.” This problem is defined as being specific for colorectal cancer but, I think we can all agree that there will not be a non-surgical effective therapy for the whole wide spectrum of any cancer type. Please replace the last part of the sentence for a more realistic statement.

Response: We thank the reviewer for the suggestion. We revised this sentence as “Colorectal cancer (CRC) is one of the most common malignancies involving a complex interplay among different cell types via multiple physical and chemical factors with relatively limited options other than surgical treatment” (page 2).

“Most previous studies focused on the generation of effective anti-tumor responses by T cells [5].”

Unclear what this refers to but I am confident that all-in-all more non-immunotherapeutic avenues have been explored, to date, than immunotherapeutic ones.

Response: We revised as “There have been many studies exploring T cells for effective anti-tumor immunotherapies” in page 3.

“In addition, infiltrating immune cells excluded TME has been shown to be associated with CRC.”

Rephrase.

Response: We revised as “It has been reported recently that exclusion of infiltrating immune cells from the TME was associated with poor prediction for CRC patients” in page 4.

Reviewer #3, expert in single cell sequencing and characterisation of the tumour microenvironment (Remarks to the Author):

This manuscript studies the tumor microenvironment of colorectal cancer (CRC), particularly one two very interesting subsets of fibroblasts and macrophages. The authors focus primarily on single cell RNA sequencing (scRNAseq) of 5 patients, profiling both the tumor as well as juxta-tumor tissue. The scRNAseq data is of high quality and the authors perform state of the art, comprehensive analyses. Hypotheses generated from this scRNAseq cohort are validated in turn with multiple public cohorts of scRNAseq and bulk RNAseq, immunohistochemistry and FACS. All of this is used to build a compelling model of the mechanisms generating these two tumor-specific non-malignant populations, the interactions between them, and their association with T cells, tumor development, and immune checkpoint inhibitor response.

The study is very well done and is appropriate for the Nature Communications audience. I have no major concerns but I detail below a number of minor suggestions that might improve the paper.

Response: We thank the reviewer for these positive comments that highlight the significance of our study.

Minor Comments:

1. English should be reviewed carefully for word choice, typos, and phrasing.
2. Term “normal” tissue should be replaced by “juxta” or “adjacent” tissue as much as possible.

Response: We thank this reviewer for the kind suggestions, and we have replaced the “normal” with “adjacent” in revised manuscript as suggested. We have reviewed carefully the manuscript for language.

3. Why are there no UMAPs for endothelial or glial markers in fig 1D?

Response: We added the features plots for endothelial (CDH5, PECAM1) and glial (S100B, CDH2) markers in revised **Fig. 1d**.

4. Throughout the manuscript there are many Kaplan Meier curves and analyses of overall survival (OS) and progression free survival (PFS). In all cases the authors use the `maxstat.test` function to split the cohort into high and low for the quantitative explanatory variable (eg CIBERSORTx estimate of cellular proportion). This is statistically acceptable but can be susceptible to small variations in the distributions. It would be nice to add in the supplemental Kaplan Meier analyses split just by median and/or the actual distributions of the explanatory variables so the reader can estimate how robust these survival analyses are with respect to the cutpoint.

Response: We thank the reviewer for this kind suggestion and we added Kaplan Meier analyses split just by median of the explanatory variables as shown in **Fig. A2**. They showed the consistent results with the `maxstat.test` function to split the cohort.

Fig. A2. Progressive free survival analyses for two groups of patients stratified by the median infiltration of FAP^+ fibroblasts (left), $MARCO^+$ macrophage, or four subgroups of patients stratified by the infiltration of both FAP^+ fibroblasts and $MARCO^+$ macrophages using Kaplan-Meier curves in TCGA CRC cohort. A two-sided log-rank test $p < 0.05$ is considered as a statistically significant difference.

5. Line 407 – “To investigate xxx” must be a typo

Response: We are sorry for this typo. We corrected it as “We investigated the alteration of myeloid cell subtypes between adjacent tissues and tumor tissues” in revised manuscript (page 12).

6. Fig 4D is difficult to gauge visually. Maybe an alternative layout, like a violin plot or the actual distributions would make it easier to estimate if the bulk of the distribution is shifted or just a few outliers.

Response: We added the actual distributions in boxplot to replace **Fig. 3d** and **Fig. 4d**.

7. Need at least some information about the cohorts for Fig 6D to gauge how well matched they are, minimally age and gender.

Response: We thank the reviewer for this suggestion. We included clinical information for patients for **Fig. 6d** in Source Data.

8. Sentence from line 502-504 seems to be incomplete.

Response: We completed this sentence as “Genes of the extracellular matrix (ECM) related pathways were highly expressed in *FAP*⁺ fibroblasts and *MARCO*⁺ macrophages, suggesting that either *FAP*⁺ fibroblasts or *MARCO*⁺ macrophages may facilitate the generation of desmoplastic structures” in the revised manuscript (page 17).

9. Fig 7A-E show a very large range for many of the boxes – the actual density and variation might be better visualized by showing the individual points instead of box and whisker.

Response: We agreed and we thus added the individual points of values to each variable.

10. Why aren't there 4 curves for fig 7H,I (*FAP*^{high/low}, *MARCO*^{high/low})?

Response: We performed Kaplan-Meier curves for four subgroups of patients with anti-PD-L1 treatment stratified by the expression of both *FAP* and *MARCO*, patients with high level of both *FAP* and *MARCO* showed the trend of the shortest survival time (**Fig. A3**).

Fig. A3. Progressive free survival analyses for four subgroups of patients stratified by the expression of both *FAP* and *MARCO* using Kaplan-Meier curves in the anti-PD-L1 immunotherapy cohort. Statistical analysis is performed by two-sided log-rank test.

References

- Guinney, J., Dienstmann, R., Wang, X., de Reynies, A., Schlicker, A., Sonesson, C., Marisa, L., Roepman, P., Nyamundanda, G., Angelino, P., *et al.* (2015). The consensus molecular subtypes of colorectal cancer. *Nat Med* *21*, 1350-1356.
- Liu, Q., Sun, Z., and Chen, L. (2020). Memory T cells: strategies for optimizing tumor immunotherapy. *Protein Cell* *11*, 549-564.
- Maglio, D.H.G., Paz, M.L., and Leoni, J. (2016). Sunlight Effects on Immune System: Is There Something Else in addition to UV-Induced Immunosuppression? *Biomed Research International* *2016*.
- Martin, J.C., Chang, C., Boschetti, G., Ungaro, R., Giri, M., Grout, J.A., Gettler, K., Chuang, L.-s., Nayar, S., Greenstein, A.J., *et al.* (2019). Single-Cell Analysis of Crohn's Disease Lesions Identifies a Pathogenic Cellular Module Associated with Resistance to Anti-TNF Therapy. *Cell* *178*, 1493-+.
- National Health Commission of the People's Republic of, C. (2020). [Chinese Protocol of Diagnosis and Treatment of Colorectal Cancer (2020 edition)]. *Zhonghua Wai Ke Za Zhi* *58*, 561-585.
- Newman, A.M., Liu, C.L., Green, M.R., Gentles, A.J., Feng, W., Xu, Y., Hoang, C.D., Diehn, M., and Alizadeh, A.A. (2015). Robust enumeration of cell subsets from tissue expression profiles. *Nat Methods* *12*, 453-457.
- Newman, A.M., Steen, C.B., Liu, C.L., Gentles, A.J., Chaudhuri, A.A., Scherer, F., Khodadoust, M.S., Esfahani, M.S., Luca, B.A., Steiner, D., *et al.* (2019). Determining cell type abundance and expression from bulk tissues with digital cytometry. *Nat Biotechnol* *37*, 773-782.
- Reiman, J.M., Knutson, K.L., and Radisky, D.C. (2010). Immune Promotion of Epithelial-mesenchymal Transition and Generation of Breast Cancer Stem Cells. *Cancer Research* *70*, 3005-3008.
- Ricciardi, M., Zanotto, M., Malpeli, G., Bassi, G., Perbellini, O., Chilosi, M., Bifari, F., and Krampera, M. (2015). Epithelial-to-mesenchymal transition (EMT) induced by inflammatory priming elicits mesenchymal stromal cell-like immune-modulatory properties in cancer cells. *British Journal of Cancer* *112*, 1067-1075.
- Witkowski, M.T., Dolgalev, I., Evensen, N.A., Ma, C., Chambers, T., Roberts, K.G., Sreeram, S., Dai, Y.L., Tikhonova, A.N., Lasry, A., *et al.* (2020). Extensive Remodeling of the Immune Microenvironment in B Cell Acute Lymphoblastic Leukemia. *Cancer Cell* *37*, 867-+.
- Zhang, L., Yu, X., Zheng, L., Zhang, Y., Li, Y., Fang, Q., Gao, R., Kang, B., Zhang, Q., Huang, J.Y., *et al.* (2018). Lineage tracking reveals dynamic relationships of T cells in colorectal cancer. *Nature* *564*, 268-+.

Reviewers' Comments:

Reviewer #1:

Remarks to the Author:

I have no more questions, congratulations on a beautiful paper.

Reviewer #2:

Remarks to the Author:

I would like to thank the authors for their efforts in addressing my comments in the previous review round. I think the readability of the manuscript has greatly improved. However, after consulting all replies to my queries I still identify a major issue that is in the way of this Reviewer having an overall positive opinion about the paper:

I disagree with the methodology employed by the authors to make use of deconvolution to determine the relative frequency of subsets across publicly available RNA sequencing cohorts. Without adequate validation this Reviewer cannot trust that CIBERSORTX is able to estimate the proportions of >50 subsets reliably. The authors assume this and I think this considers a major methodological flaw of the paper. Their validation (included in the recent version of the paper) does not answer these concerns and it has not been conducted in an adequate manner.

I urge the authors to consider whether they require CIBERSORTx to achieve the same conclusions that are now contained in the paper. If there are no other frequent cell types expressing FAP or MARCO the authors could also rely on gene expression for their correlation on bulk datasets. Instead, to demonstrate an association between the frequency of FAP+ fibroblast subsets and MARCO+ macrophages the authors can make use of their own dataset and publically available datasets. Can a correlation between the frequency of these subsets be observed there?

Other comments:

1. I still do not understand Figure 1b. How is it possible that the cell proportions remain so similar between normal and tumor? If I interpret Figure 1e. correctly, there are 4x more mast cells or plasma cells in normal samples than in tumors, it should be possible to visualize this in the UMAP embedding. It is also striking that, despite employing Harmony, the cells from normal and tumor occupy exactly the same high dimensional space. Isn't it expected that MSCs are completely different from normal and tumor? We should not see all subsets in the same proportions. As it stands, I am not confident that these results make sense or that this figure is relevant to the paper. The authors should be aware that Harmony does not distinguish batch effects from biological variation. And while it might be a useful tool to cluster and annotate cell subsets across datasets it is not ideal for detecting differences across groups. I think the authors are better off with presenting a single embedding.

2. I still disagree with the acritical employment of hallmark pathways for any cell type. The UV response pathway includes 158 genes that are involved in other processes in addition to UV response, including cell cycle and apoptosis. Since it is (very) unlikely that UV response is occurring in the colon or rectum, the authors should translate what this finding means in the context of this paper. Is there any rationale to incorporate this pathway in the category "tissue remodeling-related pathway"? In my opinion, the authors do a nice interpretation of the hit they obtained for the hypoxia pathway, for instance.

3. The introduction still lacks some clarity. Example: between lines 57-65. The authors start by saying that anti-PD-1 is only effective in the advanced setting in 5% of CRCs and that T cell based therapies have been extensively investigated. They then argue that a general lack of knowledge regarding what is happening in the transition between normal and tumor tissue is preventing the field to come up with

preventive therapeutic targets of non-T immune cells. Why preventive if the previous info was about immunotherapy in an advanced setting. And why specify non-T immune cells (very strange formulation) if the next paragraph is about mesenchymal stromal cells which are not immune?

Reviewer #3:

Remarks to the Author:

The authors have done a very nice job of responding to the initial comments and the additional characterization of spatial patterns further increases the novelty of this work.

The only substantive issue I have now concerns the choice of "MARCO+" as the name for their macrophage cluster of interest. This subpopulation of macrophages clearly exists and has interesting properties which are investigated in detail. But the evidence that MARCO in particular is highly expressed is not clear (FigS2c appears to be the only evidence, but this is impossible to see clearly). Relatedly, they show that in terms of overall gene expression profile this cluster is very similar to one labeled as "SPP1+ macrophages" in other publications. Although they don't directly show that SPP1 is specifically expressed in the "MARCO+" cluster, they have now replaced MARCO immunofluorescence from the initial manuscript with SPP1 immunofluorescence (Fig 5d). As this population is central to the study, even being mentioned specifically in the title, this deserves careful consideration. Whether MARCO, SPP1, or another gene is chosen for the annotation, it's expression pattern should be clearly depicted.

Minor comments:

Fig 1d : are panel labels correct? For example PTPRC, CD3E, and CD14 don't show expected patterns given cluster labels in Fig1b and expression values in Fig 1c.

Line 268/Fig 3H : looks like ICAM1+ (not ICAM1-) telocytes are likely source of FAP+ fibroblasts.

Fig 3 uses upper case letters (ABC, etc) for panels, but all others use lower case (abc, etc).

Close parenthesis missing on line 442.

Reviewer #4:

Remarks to the Author:

In this study, Qi and colleagues provide single cell RNA sequencing data of 5 colorectal cancer patient samples and matched normal tissues. They focus on macrophages and fibroblasts and integrate their data with existing single cell and bulk RNAseq datasets to assess associations between the presence of FAP+ fibroblasts and MARCO+ macrophages with poorer outcome of CRC patients.

While I appreciate that this study provides a very extensive bioinformatic analysis of CRC samples it mainly remains correlative and experimental evidence is missing to demonstrate what the authors claim: that FAP+ fibroblasts interact with MARCO+ macrophages and that this interaction causes a worse outcome in CRC patients.

The findings in this study are not very novel. Fibroblasts and macrophages are the most abundant non-cancerous cells in tumours, they are known to promote a poorer outcome in many cancers including CRC. Most fibroblasts express FAP and most tumour associated macrophages which are known to have rather an M2-like phenotype express MARCO (among other M2 markers). Thus, finding FAP+ and MARCO+ macrophages located next to each other in tumour tissues is not surprising, it is expected and known. At the same time this does not mean that FAP+ fibroblasts and MARCO+ macrophages are directly interacting and that the expression of FAP and/or MARCO drives a worse outcome. The authors have not experimentally proven neither a direct interaction between FAP+ fibroblasts and MARCO+ macrophages, nor a direct role for these 2 proteins (FAP and MARCO) in CRC progression. Functional biological assays to prove this are missing in this study.

In addition, therapies targeting MARCO and FAP are already being tested, so this study is not revealing new potential targets but is rather suggesting what others have already suggested and are

investigating.

I am also confused by the use of SSP1 and MARCO+ macrophages. It seems that the authors have changed MARCO for SSP1 in their read-out and in the text but the title still says MARCO+ macrophages. SSP1 and MARCO are 2 different proteins that although they might be expressed by the same cluster of macrophages they have different functions and thus it is not clear from a biological point of view which one does this study suggest should be targeted?

In Figure 4g, the text says they are analysing MARCO+ macrophages by flow cytometry but the FACS data shows CD13+ CD206+ macrophages not MARCO? The legend for this figure is not correct either.

In Figure 5D, the authors talk about co-localization of FAP+ fibroblasts and MARCO+ macrophages but the images show they have stained for SPP1 (not MARCO). Also 2 cell types cannot co-localize, they can localize close to each other within a tissue (which is actually common for fibroblasts and macrophages) but they do not co-localize. The term co-localization is used for proteins within a same cell that localize to the same spot, this would suggest that they may interact with each other but it is not a final prove. Thus, I am also confused of how the quantification of FAP+ fibroblasts co-localizing with SPP1 macrophages shown in graph 5e has been performed.

In Figure 6, the spatial transcriptomic analysis shows only 1 patient. I understand the authors had at least 5 fresh CRC patients. It is difficult to draw any conclusion from only 1 patient. The authors could have also used archived CRC samples to do spatial transcriptomics and thereby draw a more robust conclusion using more samples.

Point-by point responses to Reviewers' comments

Reviewer #1, expert in colorectal cancer genomics (Remarks to the Author):

I have no more questions, congratulations on a beautiful paper.

Response: We thank this reviewer for the kind support.

Reviewer #2, expert in colorectal cancer and immunotherapies (Remarks to the Author):

I would like to thank the authors for their efforts in addressing my comments in the previous review round. I think the readability of the manuscript has greatly improved. However, after consulting all replies to my queries I still identify a major issue that is in the way of this Reviewer having an overall positive opinion about the paper:

I disagree with the methodology employed by the authors to make use of deconvolution to determine the relative frequency of subsets across publicly available RNA sequencing cohorts. Without adequate validation this Reviewer cannot trust that CIBERSORTX is able to estimate the proportions of >50 subsets reliably. The authors assume this and I think this considers a major methodological flaw of the paper. Their validation (included in the recent version of the paper) does not answer these concerns and it has not been conducted in an adequate manner.

Response: As mentioned in the previous response letter, CIBERSORTx allows better building of custom signature matrixes from single-cell or flow-sorted bulk transcriptomic data to reconstruct cell-specific transcriptional profiles¹. CIBERSORTx package has been proven to be a powerful tool in the field to predict the cell composition from bulk RNA sequencing data, which would reveal the possibilities of candidate cell populations^{2,3}. We only used it to find the clues or correlation between the infiltration of *FAP*⁺ fibroblasts and *MACRO*⁺ macrophages estimated by CIBERSORTx, which we further thoroughly studied to reveal their interaction in tumor experimentally by immunofluorescence and spatial transcriptomics.

We agree with this reviewer that it may be difficulty for CIBERSORTx to estimate larger than 50 subsets, therefore, we additionally assessed the performance of CIBERSORTx to estimate all cell types using our own scRNA-seq data. To further evaluate the robustness of CIBERSORTx in estimating cell proportion of more than 50 subsets, we created a signature matrix from a training cohort consisting of two CRC specimens in our scRNA-seq dataset and used the remaining three CRC specimens to make a pseudo-bulk transcriptomics as the test dataset. We next tested the accuracy of CIBERSORTx in estimating the cell proportion by comparing estimated percentile with real percentile based on the original dataset. The results show great performance using deconvolution results based on created signature matrix which distinguished all cell types or subtypes of each major types (**Fig. 1 for reviewer**)

Although we show the good performance of CIBERSORTx in estimating cell percentage, nerve the less, we still agree that this only provide the clue that the *FAP*⁺ fibroblasts and *MACRO*⁺ macrophages were highly correlated. We further confirmed our hypothesis using other strategies, including immunofluorescence staining and

spatial transcriptomic profiling.

Fig. 1 for reviewer. Robustness of CIBERSORTx in dissection of solid tumors using single-cell reference profiles. (A-F) Correlation of cell subset proportions of 58 cell types (A), subtypes of MSCs (B), myeloid cells (C), endothelial cells (D), B cells (E), and T/ILC cells (F) measured by CIBERSORTx deconvolution, versus the ground truth cell proportions in scRNA-seq data generated in this paper.

I urge the authors to consider whether they require CIBERSORTx to achieve the same conclusions that are now contained in the paper. If there are no other frequent cell types expressing *FAP* or *MARCO* the authors could also rely on gene expression for their correlation on bulk datasets. Instead, to demonstrate an association between the frequency of *FAP*⁺ fibroblast subsets and *MARCO*⁺ macrophages the authors can make use of their own dataset and publically available datasets. Can a correlation between the frequency of these subsets be observed there?

Response: Thanks for the reviewer's great suggestion. Indeed, the *FAP* and *MARCO* mainly expressed in *FAP*⁺ fibroblast and *MARCO*⁺ macrophages. The expression of *FAP* and *MARCO* indeed showed significant correlation (**Fig. 2 for reviewer**).

As *SPPI* and *MARCO* labeled the same macrophage cluster, we also showed that *FAP* and *SPPI* expression were highly correlated (**Fig. 3 for reviewer, and see comment from Reviewer # 3**).

Fig. 2 for reviewer. The spearman correlation of gene expression between *MARCO* and *FAP*.

Fig. 3 for reviewer. The spearman correlation of gene expression between *SPP1* and *FAP*.

Other comments:

1. I still do not understand Figure 1b. How is it possible that the cell proportions remain so similar between normal and tumor? If I interpret Figure 1e. correctly, there are 4x more mast cells or plasma cells in normal samples than in tumors, it should be possible to visualize this in the UMAP embedding. It is also striking that, despite employing Harmony, the cells from normal and tumor occupy exactly the same high

dimensional space. Isn't it expected that MSCs are completely different from normal and tumor? We should not see all subsets in the same proportions. As it stands, I am not confident that these results make sense or that this figure is relevant to the paper. The authors should be aware that Harmony does not distinguish batch effects from biological variation. And while it might be a useful tool to cluster and annotate cell subsets across datasets it is not ideal for detecting differences across groups. I think the authors are better off with presenting a single embedding.

Response: We are sorry for this unclarity. In our previous version, the dot indicating the size of each cell is a bit large, and each group contains many cells, thus making the adjacent normal tissue and tumor tissue appear similar in visualization. We grouped the adjacent normal and tumor tissues on UMAP, as **Fig. 4** for reviewer shown, the cell proportions are significant different between the adjacent normal tissue and tumor tissue. The difference of each cell type between the adjacent normal tissue and tumor tissue is consistent with the clustering of each major type (**Fig. 4** for reviewer, **Fig. S2e-h, 3a, and 4a**).

Fig. 4 for reviewer. The distribution of cell types between adjacent normal and tumor tissues. (A) The UMAP shows the distribution of adjacent normal tissue (blue) and tumor tissue (orange red) in all cells. **(B)** UMAP of the composition of all 58 cell types in adjacent normal and tumor tissues. Each dot represents one cell and the color indicates the cell type.

2. I still disagree with the acritical employment of hallmark pathways for any cell type. The UV response pathway includes 158 genes that are involved in other processes in addition to UV response, including cell cycle and apoptosis. Since it is (very) unlikely that UV response is occurring in the colon or rectum, the authors should translate what this finding means in the context of this paper. Is there any rationale to incorporate this pathway in the category “tissue remodeling-related pathway”? In my opinion, the authors do a nice interpretation of the hit they obtained for the hypoxia pathway, for instance.

Response: The categories are provided by Molecular Signatures Database (MSigDB) database, and we compared the enrichment score of these gene sets across individual cell types between tumors and adjacent normal tissues. We agree with reviewer’s comment that UV response is unlikely occurred in the colon or rectum, and as the reviewer’s suggested, the UV response pathway includes genes involved in other process, we found that 62 genes and 33 genes in the above annotated UV response pathway are involved in cell cycle and apoptosis, respectively. To avoid the acritical employment of hallmark pathways, we have kept gene sets in the immune, metabolism, signaling, and proliferation that may be involved in the regulation of CRC in the revised **Fig. 2a**.

3. The introduction still lacks some clarity. Example: between lines 57-65. The authors start by saying that anti-PD-1 is only effective in the advanced setting in 5% of CRCs and that T cell based therapies have been extensively investigated. They then argue that a general lack of knowledge regarding what is happening in the transition between normal and tumor tissue is preventing the field to come up with preventive therapeutic targets of non-T immune cells. Why preventive if the previous info was about immunotherapy in an advanced setting. And why specify non-T immune cells (very strange formulation) if the next paragraph is about mesenchymal stromal cells which are not immune?

Response: To make the introduction part more read-able, we re-vised this paragraph as follows: “There have been many studies exploring T cells for effective anti-tumor immunotherapies⁴. However, the PD-1-targeting antibody, pembrolizumab, is only effective for mismatch repair-deficient tumors with high microsatellite instability (MSI-H), which account for less than 5% of metastatic CRC cases^{5,6}. Therefore, it is necessary to understand the mechanism of cellular and molecular remodeling in the tumor microenvironment (TME) of CRC, and find potential intervention targets to enhance the therapeutic efficacy of immunotherapy. Recent research has revealed that

stromal cells and myeloid cells may form a distinctive niche for tumor growth, metastasis, and immune exclude microenvironment^{7,8}, making them potential therapeutic targets”.

Reviewer #3, expert in single cell sequencing/TME (Remarks to the Author):

The authors have done a very nice job of responding to the initial comments and the additional characterization of spatial patterns further increases the novelty of this work.

The only substantive issue I have now concerns the choice of “MARCO+” as the name for their macrophage cluster of interest. This subpopulation of macrophages clearly exists and has interesting properties which are investigated in detail. But the evidence that MARCO in particular is highly expressed is not clear (FigS2c appears to be the only evidence, but this is impossible to see clearly). Relatedly, they show that in terms of overall gene expression profile this cluster is very similar to one labeled as “SPP1+ macrophages” in other publications. Although they don’t directly show that SPP1 is specifically expressed in the “MARCO+” cluster, they have now replaced MARCO immunofluorescence from the initial manuscript with SPP1 immunofluorescence (Fig 5d). As this population is central to the study, even being mentioned specifically in the title, this deserves careful consideration. Whether MARCO, SPP1, or another gene is chosen for the annotation, it’s expression pattern should be clearly depicted.

Response: We thank this reviewer for the kind suggestion. After carefully comparing our data to that from Zhang *et al. Cell* 2020⁹ and Lee *et al. Nature Genetics* 2020¹⁰ (Fig. S5), we agree that the *MARCO*⁺ macrophages were the same cluster as “*SPP1*⁺ macrophages” (Fig. 5 for reviewer). We would happy to change the annotation of “*MARCO*⁺ macrophages” to “*SPP1*⁺ macrophages” in our revision.

Fig. 5 for reviewer. The expression distribution of *SPP1* and *MARCO* among the different subtypes of myeloid cells.

Minor comments:

Fig 1d : are panel labels correct? For example PTPRC, CD3E, and CD14 don't show expected patterns given cluster labels in Fig1b and expression values in Fig 1c.

Response: We are sorry for the mismatch between the figure and label, and we have corrected it in **Fig.1d**.

Line 268/Fig 3H: looks like ICAM1+ (not ICAM1-) telocytes are likely source of FAP+ fibroblasts.

Response: Sorry for this typo, we have corrected it.

Fig 3 uses upper case letters (ABC, etc) for panels, but all others use lower case (abc, etc).

Close parenthesis missing on line 442.

Response: We have made the consistent lowercase letters in all figures.

Reviewer #4, expert in CAF/TAM cells (Remarks to the Author):

In this study, Qi and colleagues provide single cell RNA sequencing data of 5 colorectal cancer patient samples and matched normal tissues. They focus on macrophages and fibroblasts and integrate their data with existing single cell and bulk RNAseq datasets to assess associations between the presence of FAP+ fibroblasts and MARCO+ macrophages with poorer outcome of CRC patients.

While I appreciate that this study provides a very extensive bioinformatic analysis of CRC samples it mainly remains correlative and experimental evidence is missing to demonstrate what the authors claim: that FAP+ fibroblasts interact with MARCO+ macrophages and that this interaction causes a worse outcome in CRC patients.

Response: We thank this reviewer for his/her appreciation of “this study provides a very extensive bioinformatic analysis of CRC samples”. In the final revised manuscript, we have provided additional evidence showing that *FAP*⁺ fibroblasts interact with *SPP1*⁺ macrophages as described below.

- 1) We experimentally prove that *FAP*⁺ fibroblasts and *SPP1*⁺ macrophages were localized closely by immuno-fluorescent staining and spatial transcriptomics. In our revised manuscript, we added ST datasets up to four patients.
- 2) We assessed the protein expression levels by the H-score system in tissue microarray (TMA) of 78 CRC patients to investigate whether FAP and SPP1 was specifically increased in tumor tissue compared with adjacent normal tissue, and indeed, we found patients with high protein level of FAP or SPP1 enrolled in this TMA cohort exhibited shorter OS. Furthermore, patients with high protein levels of both FAP and SPP1 survived poorly compared to patients with lower level of FAP or SPP1 (**Fig. 6 for reviewer**).

Fig. 6 for reviewer. Tissue microarray analysis. (A-B) Quantification of FAP (A) and SPP1 (B) staining intensity in the adjacent normal and CRC tissue microarray. (C-E) Kaplan-Meier curves showed overall survival analyses for low and high expression FAP alone (C), SPP1 alone (D), and both (E) CRC patients. A paired two-sided paired Student’s t-test was used to assess the difference in A-B. A two-sided log-rank test was used to assess statistical significance in C-E.

Beside immunofluorescent staining, spatial transcriptomics, and TMA data, we also demonstrated that *FAP*⁺ fibroblasts and *SPP1*⁺ macrophages co-enrichment in the tumor and high infiltration of these two subsets in scRNA-seq dataset, 15 public bulk

transcriptomics dataset and ICB treated dataset. We also identified potential biologically active mediators between these two subsets. These results suggested high probability of the interaction between *FAP*⁺ fibroblasts and *SPPI*⁺ macrophages.

The reviewer suggested more functional assay to demonstrate the biological impact of direct interaction between these two cell types, we agree that it is indeed very important to do but we believe that these future experiments are beyond the scope of current study. However, we have added discussion of these potential future experiments in revision (page 23 in manuscript).

The findings in this study are not very novel. Fibroblasts and macrophages are the most abundant non-cancerous cells in tumours, they are known to promote a poorer outcome in many cancers including CRC.

Response: We agree that only the abundance of fibroblasts and macrophages is not very novel but our study is about the novel interaction of two different subsets of fibroblasts and macrophages for their critical roles in TME.

1) This study characterized the cell composition alteration of TME between tumor and to adjacent normal tissues base on the scRNA-seq data, but not the total of fibroblasts and macrophages. *FAP*⁺ fibroblasts and *SPPI*⁺ macrophages were increased in tumor, while we also show that not all subsets of fibroblasts and macrophages are increased in tumor, the infiltration of *NT5E*⁺ fibroblasts, *FGFR2*⁺ fibroblasts, and *MFAP5*⁺ myofibroblasts were decreased in tumor compared to adjacent normal tissues (**Fig. S4b**). *CD24*⁺ fibroblasts, *FGFR2*⁺ fibroblasts, *CIQC*⁺ *MRC1*⁻ macrophages, *VCAN*⁺ macrophages, *THBS1*⁺ macrophages were non-significant alteration between tumor and to adjacent normal tissues (**Fig. S4b and S6b**).

2) Among increased subsets, *FAP*⁺ fibroblasts and *SPPI*⁺ macrophages (not others) co-appear in the tumor were identified in scRNA-seq and 15 CRC transcriptomics datasets by bioinformatic analysis, and validated in IF and ST datasets. Furthermore, high infiltration of both these two subsets is associated with worse survival but not the other subsets.

3) Most importantly, these two subsets localize closely to exclude the immune signature of the tumor, with an important implication in anti-tumor immunotherapy (**Fig. 6 and 8a-i**).

4) In addition, there are multiple active mediators (ligand-receptors pairs) between these two interacting cells (**Fig.7**), which may be potential targets for disrupting the interaction of these two subsets in future study.

Most fibroblasts express FAP and most tumour associated macrophages which are known to have rather an M2-like phenotype express MARCO (among other M2 markers). Thus, finding *FAP*⁺ and *MARCO*⁺ macrophages located next to each other in tumour tissues is not surprising, it is expected and known.

Response: We respectively disagree with this. Based on our data and that from others,

FAP significantly enriched in *FAP*⁺ fibroblasts which represented about 44% of total stromal cells (**Fig. 3a-c, 3f**), and *SPP1/MARCO* significantly enriched *SPP1*⁺ macrophages which only represented 11.6 % of TAMs (**Fig. 4a-c, 4f, and Fig. 5 for reviewer**). As we mentioned above, the diversity as putative interaction of cell populations of fibroblasts and macrophages in TME were highly complex, and our analysis and experiments reveal the closely located unique subsets of fibroblasts and macrophages in TME. We also provide critical information about the consequences of such interaction and it is clearly important to target the interaction between these two cell subsets to boost anti-tumor immunotherapy, which has not been proposed before.

At the same time this does not mean that *FAP*⁺ fibroblasts and *MARCO*⁺ macrophages are directly interacting and that the expression of *FAP* and/or *MARCO* drives a worse outcome.

The authors have not experimentally proven neither a direct interaction between *FAP*⁺ fibroblasts and *MARCO*⁺ macrophages, nor a direct role for these 2 proteins (*FAP* and *MARCO*) in CRC progression. Functional biological assays to prove this are missing in this study.

Response: We provide strong evidences to show that they interact in the tumor and their expression led to a worse outcome in cancer patients, using methods such as IF, ST, TMA analysis, and the ligand-receptors interaction between these subsets as above mentioned. The functional assay to prove that the direct interaction in anti-tumor therapy is indeed very important, but we believe is beyond the scope of current study.

In addition, therapies targeting *MARCO* and *FAP* are already being tested, so this study is not revealing new potential targets but is rather suggesting what others have already suggested and are investigating.

Response: We thank the reviewer to point out this issue. Therapies targeting *MARCO* and *FAP* were developed in mouse model¹¹⁻¹⁴, but not successful in clinical trials for cancer therapy strategies. There could be many possibilities given the complex TME in cancer patients. This seems also in-line with our proposal that we may have to disrupt the interaction of these two subsets of tumor infiltrating fibroblasts and macrophages to gain better therapy. Thus we would propose to target their interaction but individual subset. Our study is novel not only for the findings but also our concept. Our dataset could provide more comprehensive and precise functional and regulatory image of these two molecule targeted cell types for future drug development of antibody-drug-conjugate (ADC) and bi-specific antibodies to better perturbing these two cell types or even blocking their interactions. Therefore, our study is actually providing new potential targets on these two cell types rather than only targeting on *MARCO* or *FAP*.

I am also confused by the use of *SSP1* and *MARCO*⁺ macrophages. It seems that the

authors have changed MARCO for SSP1 in their read-out and in the text but the title still says MARCO+ macrophages. SSP1 and MARCO are 2 different proteins that although they might be expressed by the same cluster of macrophages they have different functions and thus it is not clear from a biological point of view which one does this study suggest should be targeted?

Response: We are sorry for this unclarity. As we respond to reviewer #3's comment above, the subset of *MARCO*⁺ macrophages were indeed the same subset as *SPP1*⁺ macrophages. We revised our manuscript accordingly using SPP1 macrophages.

In Figure 4g, the text says they are analysing MARCO+ macrophages by flow cytometry but the FACS data shows CD13+ CD206+ macrophages not MARCO? The legend for this figure is not correct either.

Response: Due to technical issues, we were unable to stain MARCO+ population in tumor. We used other surrogate surface markers to stain this population from our scRNA-seq analyses including *ANPEP* (encoding CD13), and *MRC1* (encoding CD206) since *MARCO*⁺/*SPP1*⁺ macrophages were positive for these two markers. Considering the previous publication showing that CD13 was used to label a unique subset of tumor associated myeloid cells¹⁵, we used the combination of CD45⁺ CD3⁻ CD19⁻ XCR1⁻ CD1c⁻ CD16B⁻ CD14⁺ CD206⁺ CD13⁺ to mark the *MARCO*⁺ macrophages population. The CD13⁺ CD206⁺ macrophages might include some *VCAN*⁺ macrophages but these cells were not changed in tumor compared to normal tissue according to scRNA-seq data.

In Figure 5D, the authors talk about co-localization of FAP+ fibroblasts and MARCO+ macrophages but the images show they have stained for SPP1 (not MARCO).

Response: Sorry for this confusion again and as shown in the response above, we have renamed *MARCO*⁺ macrophages to "*SPP1*⁺ macrophages". *SPP1* could be used as a biomarker of *SPP1*⁺ macrophages and there is a commercially available anti-SPP1 antibody working well in immunofluorescence assay.

Also 2 cell types cannot co-localize, they can localize close to each other within a tissue (which is actually common for fibroblasts and macrophages) but they do not co-localize. The term co-localization is used for proteins within a same cell that localize to the same spot, this would suggest that they may interact with each other but it is not a final prove. Thus, I am also confused of how the quantification of FAP+ fibroblasts co-localizing with SPP1 macrophages shown in graph 5e has been performed.

Response: We have corrected it as "localize closely" in the revised manuscript and Fig. 5e.

In Figure 6, the spatial transcriptomic analysis shows only 1 patient. I understand the

authors had at least 5 fresh CRC patients. It is difficult to draw any conclusion from only 1 patient. The authors could have also used archived CRC samples to do spatial transcriptomics and thereby draw a more robust conclusion using more samples.

Response: We appreciate the suggestions from the reviewer. Besides the two patients enrolled in the previous version (#P6, P7; **Fig. 6 and S8**), we also performed two additional ST from CRC patients (**Fig. S8f-k, Fig. S9**). The new datasets together with the previous two patients showed consistent results that *SPPI*⁺ macrophages and *FAP*⁺ fibroblasts can be located on the same spots and wrapped around malignant epithelial cells, and the features indicating infiltrating T cells or B cells were excluded from tumor core.

References

- 1 Newman, A. M. *et al.* Determining cell type abundance and expression from bulk tissues with digital cytometry. *Nat Biotechnol* **37**, 773-782, doi:10.1038/s41587-019-0114-2 (2019).
- 2 Luca, B. A. *et al.* Atlas of clinically distinct cell states and ecosystems across human solid tumors. *Cell* **184**, 5482-5496 e5428, doi:10.1016/j.cell.2021.09.014 (2021).
- 3 Steen, C. B. *et al.* The landscape of tumor cell states and ecosystems in diffuse large B cell lymphoma. *Cancer Cell* **39**, 1422-1437 e1410, doi:10.1016/j.ccell.2021.08.011 (2021).
- 4 Egen, J. G., Ouyang, W. & Wu, L. C. Human Anti-tumor Immunity: Insights from Immunotherapy Clinical Trials. *Immunity* **52**, 36-54, doi:10.1016/j.immuni.2019.12.010 (2020).
- 5 Nordholm-Carstensen, A., Krarup, P. M., Morton, D., Harling, H. & Danish Colorectal Cancer, G. Mismatch repair status and synchronous metastases in colorectal cancer: A nationwide cohort study. *Int J Cancer* **137**, 2139-2148, doi:10.1002/ijc.29585 (2015).
- 6 Le, D. T. *et al.* PD-1 Blockade in Tumors with Mismatch-Repair Deficiency. *N Engl J Med* **372**, 2509-2520, doi:10.1056/NEJMoal1500596 (2015).
- 7 Hegde, P. S. & Chen, D. S. Top 10 Challenges in Cancer Immunotherapy. *Immunity* **52**, 17-35, doi:10.1016/j.immuni.2019.12.011 (2020).
- 8 Binnewies, M. *et al.* Understanding the tumor immune microenvironment (TIME) for effective therapy. *Nat Med* **24**, 541-550, doi:10.1038/s41591-018-0014-x (2018).
- 9 Zhang, L. *et al.* Single-Cell Analyses Inform Mechanisms of Myeloid-Targeted Therapies in Colon Cancer. *Cell* **181**, 442-459 e429, doi:10.1016/j.cell.2020.03.048 (2020).
- 10 Lee, H. O. *et al.* Lineage-dependent gene expression programs influence the immune landscape of colorectal cancer. *Nature Genetics* **52**, 594+, doi:10.1038/s41588-020-0636-z (2020).
- 11 La Fleur, L. *et al.* Expression of scavenger receptor MARCO defines a targetable tumor-associated macrophage subset in non-small cell lung cancer.

- Int J Cancer* **143**, 1741-1752, doi:10.1002/ijc.31545 (2018).
- 12 Busek, P., Mateu, R., Zubal, M., Kotackova, L. & Sedo, A. Targeting fibroblast activation protein in cancer - Prospects and caveats. *Front Biosci (Landmark Ed)* **23**, 1933-1968, doi:10.2741/4682 (2018).
- 13 Yang, M. *et al.* Inhibition of MARCO ameliorates silica-induced pulmonary fibrosis by regulating epithelial-mesenchymal transition. *Toxicol Lett* **301**, 64-72, doi:10.1016/j.toxlet.2018.10.031 (2019).
- 14 Feig, C. *et al.* Targeting CXCL12 from FAP-expressing carcinoma-associated fibroblasts synergizes with anti-PD-L1 immunotherapy in pancreatic cancer. *Proc Natl Acad Sci U S A* **110**, 20212-20217, doi:10.1073/pnas.1320318110 (2013).
- 15 Dondossola, E. *et al.* CD13-positive bone marrow-derived myeloid cells promote angiogenesis, tumor growth, and metastasis. *Proc Natl Acad Sci U S A* **110**, 20717-20722, doi:10.1073/pnas.1321139110 (2013).

Reviewers' Comments:

Reviewer #2:

Remarks to the Author:

I thank the authors for addressing my previous comments. I have one final suggestion: I think the UMAP graph that was provided for review is much more clear than the one that is currently included in Figure 1 (in order to compare tumor and normal). I would favor the replacement of the current graph.

Reviewer #3:

Remarks to the Author:

I am satisfied with the response to the last round of comments, the manuscript looks ready to publish as far as I am concerned. Nice job.

Reviewer #4:

Remarks to the Author:

The authors have addressed most of my questions but I have one more question concerning the new data presented in Figure 6 in the rebuttal letter. The survival blue curve in E is the same as in C and the red curve in E is the same as in D. This would mean that there is a perfect match and that all FAP low patients are also SPP1 low and all SPP1 high are also FAP high. Is that the case? are all SPP1+ samples all FAP+ ? There are 0 patients who express high levels of SPP1 but low FAP? Could you please clarify/confirm this? It would also be good to show some stainings of this TMA and include these data in the manuscript.

REVIEWERS' COMMENTS

Reviewer #2 (Remarks to the Author):

I thank the authors for addressing my previous comments. I have one final suggestion: I think the UMAP graph that was provided for review is much more clear than the one that is currently included in Figure 1 (in order to compare tumor and normal). I would favor the replacement of the current graph.

Response: We thank the reviewer's suggestion, we have replaced the updated UMAP graph to make it more clear.

Reviewer #3 (Remarks to the Author):

I am satisfied with the response to the last round of comments, the manuscript looks ready to publish as far as I am concerned. Nice job.

Response: We appreciate the reviewer's careful evaluation of our manuscript.

Reviewer #4 (Remarks to the Author):

The authors have addressed most of my questions but I have one more question concerning the new data presented in Figure 6 in the rebuttal letter. The survival blue curve in E is the same as in C and the red curve in E is the same as in D. This would mean that there is a perfect match and that all FAP low patients are also SPP1 low and all SPP1 high are also FAP high. Is that the case? are all SPP1+ samples all FAP+ ? There are 0 patients who express high levels of SPP1 but low FAP? Could you please clarify/confirm this? It would also be good to show some stainings of this TMA and include these data in the manuscript.

Response: We thank the reviewer's kind support. In the **Figure 6 for reviewer**, the result showed all patients with high SPP1 have high FAP in tissue microarray with 78 colorectal cancers. The result of **Figure 8f-h** in RNA expression level also showed that most patients (60 out of 64) with high SPP1 have high expression of FAP. These results support our conclusion that patients with high SPP1 tend to have high FAP. None of the patients here expressed high levels of SPP1 but low FAP, probably due to our relatively small sample size of only 78 individuals. We add this section of the results to **Figure. 5d-h** and discuss it in the revised manuscript. In addition, we added raw figures of TMA staining in **source data files**.